# Evaluation of Global Fire Weather Database re-analysis and short-term forecast products

Robert D. Field[1,2]

[1]Department of Applied Physics and Applied Mathematics, Columbia University, New York, 10025, USA
[2]NASA Goddard Institute for Space Studies, New York, 10025, USA

*Correspondence to*: Robert D. Field (robert.field@columbia.edu)

**Abstract.**

Daily Fire Weather Index (FWI) System components calculated from the NASA Modern-Era Retrospective Analysis for Research and Applications version 2 (MERRA2) are compared to FWI calculations from a global network of
weather stations over 2004-2018, and short-term, experimental (8-day) daily FWI forecasts are evaluated for their skill across the Terrestrial Ecoregions of the World for 2018. FWI components from MERRA2 were, in general, biased low compared to station data, but this reflects a mix of coherent low and high biases of different magnitudes. Biases in different MERRA2 FWI components were related to different biases in weather input variables for different regions, but temperature and relative humidity biases were the most important overall. FWI forecasts had high skill for 1-2 day
lead times for most of the world. For longer lead-times, forecast skill decreased most quickly at high latitudes, and was most closely related to decreasing skill of relative humidity forecasts. These results provide a baseline for the evaluation and use of fire weather products calculated from global analysis and forecast fields.

## 1 Introduction

The Fire Weather Index (FWI) System is most commonly used fire danger rating system around the world (de Groot
and Flannigan, 2014;de Groot et al., 2015). It is composed of three moisture codes that track the moisture content of litter and forest floor moisture content, and three fire behaviour indices which capture potential fire spread, fuel consumption and intensity. All codes and indices are relative (unitless) measures and are interpreted differently in different fire environments. FWI calculations require 12:00 local time 2m temperature and relative humidity, 10m wind speed, and 24-hour precipitation. Snow depth is also needed in cold regions to start and stop the FWI calculations.
Because each day's calculation requires the previous day's moisture codes, weather records must be continuous and any missing data must be estimated (Lawson and Armitage 2008; Taylor and Alexander 2006). Too much missing weather data, can lead to errors that accumulate over time.

The Fine Fuel Moisture Code (FFMC) captures changes in the moisture content of fine fuels and leaf litter on the
forest floor, where fires can most easily start, and is calculated using temperature, relative humidity, precipitation and wind speed as inputs. The Duff Moisture Code (DMC) captures the moisture content of loosely compacted forest floor organic matter, and the moisture content of dead, medium-size fuels on the forest floor. The DMC is calculated from temperature, relative humidity and precipitation. The Drought Code (DC) captures the moisture content of deep,

compacted organic soils and heavy surface fuels, and is calculated from temperature and precipitation. The three
moisture codes are calculated on a daily basis using the previous day's moisture codes and the current day's weather.
Each has a precipitation threshold below which small amounts of precipitation have no effect on the code, which are
0.5 mm for the FFMC, 1.5 mm for the DMC, and 2.8 mm for the DC. The three fire behaviour indices reflect the
behaviour of a fire if it were to start. The Initial Spread Index (ISI) is driven by wind speed and FFMC and represents
the ability of a fire to spread immediately after ignition. The Buildup Index (BUI) is calculated from the DMC and
DC and represents the total fuel available to burn. The Fire Weather Index (FWI) combines the ISI and BUI to provide
an overall measure of fire danger. For all moisture codes and fire behaviour indices, increasing values indicate
decreasing moisture content. Technical details of the FWI System can be found in various technical reports (Dowdy
et al., 2009;Van Wagner, 1987), and the equation source code through publicly available repositories (Cantin, 2016).

A representative set of FWI adaptation approaches for different fire environments is listed in Table 1. When introduced
into a new region, the FWI System is calibrated for local conditions, usually with FWI calculated from weather station
data, but assembling the continuous hourly weather records needed for FWI calculations can be hard. To that end, the
Global Fire Weather Database (GFWED) provides different global FWI datasests using a combination of
meteorological reanalysis and forecasts, and precipitation estimates from rain gauges and satellites. The first iteration
of GFWED was based on the original NASA Modern-Era Retrospective Analysis for Research and Applications
(MERRA) reanalysis (Rienecker et al.), with two alternative versions substituting reanalysis precipitation for two
gridded rain-gauge products (Sheffield et al., 2006;Chen et al., 2008). These three versions were evaluated by
examining differences only between the Drought Code computed from weather stations and from gridded
meteorological products for a small number of weather stations in representative fire environments (Field et al., 2015).

Since then, a number of different versions have been added. The current 'historical' versions are based on MERRA
version 2 (Gelaro et al., 2017) since 1980, which has replaced MERRA. For the purposes of FWI calculations, the
main changes in MERRA2 from MERRA are precipitation related. As described in Gelaro et al. (2017), these include
changes to condensate re-evaporation processes and to the deep convection parameterization in the underlying model,
assimilation of additional microwave and infrared radiances from satellite along with the omission of others to which
precipitation was too sensitive, and separate constraints in the analysis adjustments on conversation of total dry
atmospheric mass and total changes in atmospheric water being equal to the net source of water from precipitation
and surface evaporation. Globally, MERRA2 has a high precipitation bias relative to GPCP (similar to other
reanalysis), but is improved from MERRA in that an apparently spurious increase from 2000 to 2010 is now absent.
During boreal summer, there is an increase in positive precipitation bias over northern Canada and northern Eurasia.
A strong negative precipitation bias in MERRA2 over much of South America has been reduced, and regional positive
and negative biases over Africa are similar. For the US during the summer where more detailed evaluation has been
done, there is improvement in MERRA2 primarily in the interannual variability in regional precipitation and in high
rainfall events compared to MERRA.

There are also near-real time 8-day FWI forecasts using weather inputs from the NASA Goddard Earth Observing System Version 5 (GEOS-5) (Rienecker et al., 2008;Molod et al., 2015), which is also the numerical weather prediction model underlying MERRA2. For both the MERRA2 reanalysis and GEOS-5 near-real time products, there are alternative precipitation versions using Global Precipitation Climatology Project One Degree Daily product (Huffman et al., 2001) since 1997, the Tropical Rainfall Measuring Mission (TRMM) (Huffman et al., 2017) Multi-satellite Precipitation Analysis (TMPA) 3B42 daily product over 1998-2015, and the Integrated Multi-satellitE Retrievals for Global Precipitation Measurement (IMERG) mission (Hou et al., 2014;Skofronick-Jackson et al., 2017) products since mid-2014. The time periods, coverage and resolution of all products are summarized at https://data.giss.nasa.gov/impacts/gfwed/.

A comparison of FWI calculated from MERRA2, rain gauge and satellite precipitation estimates was completed for a series of recent fire seasons in Canada, Chile, Greece and Indonesia (Field, in press). The focus of this paper is strictly on the evaluation of the MERRA2 reanalysis globally and over a longer period, and of the GEOS-5 8-day forecast FWI for a single year. The first goal is to compare all FWI System components (and not only the DC) calculated over a global weather station network to FWI fields calculated from MERRA2, and to understand how biases in the MERRA2 FWI are related to biases in different weather inputs. This follows comparisons of FWI computed from Weather Research and Forecasting (WRF) high-resolution analysis fields to station data over New Zealand (Simpson et al., 2014) and the McArthur Forest Fire Danger Index over Australia (Clarke et al., 2013), comparisons of FWI computed from station data and three reanalyses over the Iberian Peninsula (Bedia et al., 2012), comparisons of FWI computed from station data to high-resolution analyses over the US Great Lakes region (Horel et al., 2014), and a first global comparison of FWI computed from station data to ERA-Interim reanalyses (Vitolo et al., 2019).

The second goal is to evaluate the skill of experimental, short-term (8-day) FWI System forecasts computed from NASA GEOS-5 weather forecasts. The basic question here is: over different regions, how does fire weather forecast skill deteriorate at lead times of up to 8 days? This follows previous work to evaluate FWI from analyses for predicting global burned area (Di Giuseppe et al., 2016), smoke emissions for chemical weather forecasting for 3 months in 2013 (Di Giuseppe et al., 2018), 5 months in 2015 (Di Giuseppe et al., 2017), 5-day WRF forecasts of FWI and National Fire Danger Rating System components over Alaska in 2005 (Mölders, 2010), and 24h and 48h FWI forecasts over the US Great Lakes Region (Horel et al., 2014) for April to September 2012. The evaluation here is limited to the skill of the GEOS-5 FWI forecasts compared to FWI analyses, and not their skill in predicting fire activity or behaviour.

## 2 Data and Methods

FWI fields are computed from NASA MERRA2 reanalysis and GEOS-5 forecasts using the same approach described in Field et al. (2015). The exception is that unvegetated areas have been masked out using the GlobCover 2009 land cover classification (Arino et al., 2012), rather than annual mean temperature and precipitation thresholds. Weather station data was obtained from the National Oceanographic and Atmospheric Administration's National Center for Environmental Information (NCEI) Integrated Surface Database (ISD) of hourly and synoptic-frequency weather data

(Smith et al., 2011). As of 2019, there are 29 780 uniquely-identified stations in the ISD, but many have long periods of missing data, or report only for a short time. To strike a balance between data completeness and coverage, stations were selected that had at least 90 hourly observations for least 90% of months over 2004-2018. This initial filter only considers monthly observation counts, and not their diurnal representativeness, whether the individual FWI weather input values are reported, or whether those values passed NCEI quality control.

Hourly weather values were interpolated linearly from synoptic values, after excluding observations flagged by the NCEI as suspect or erroneous. Local 12:00 values were extracted from the interpolated hourly data with the requirement that there be actual observations within three hours before and three hours after 12:00 local time, so that 12:00 estimates were not overly influenced by observations too early or too late in the day. Precipitation was totalled from 6, 12, 18 and 24-hour reports. Snow depth from ISD reports was supplemented with data from the Global Historical Climate Network (GHCN). For many stations, snow depth from ISD and GHCN is missing during the summer, rather than reported as 0. Non-reporting snow during summer was distinguished from stations where no snow occurs using the daily Aqua MODIS snow cover fraction MYD10C1 product (Hall and Riggs, 2016). Remaining missing temperature, relative humidity and windspeed values for FWI calculations were sampled from MERRA2 fields at each station's location for the sake of continuing calculations. Missing 24-hour precipitation was taken from the CPC gridded daily precipitation estimate (Chen et al., 2008). No bias correction was applied to these filled weather input values, but there was a further requirement that FWI component values for a given day were only included in calculating bias and correlations with MERRA2 and station FWI if RH (and therefore T also) were filled from MERRA2 for no more than 20% of days during the previous 60 days.

Station-based calculations were again filtered for completeness, with the requirement that at least 80% of temperature, relative humidity and windspeed values be from observations rather than sampled from MERRA2, and that 50% of precipitation values be from observations rather than CPC, following filtering for the AgMERRA product (Ruane et al., 2014). After this requirement, there were 1746 stations (Figure 1), shown with the standard Global Fire Emissions Database (GFED) (van der Werf et al., 2017) regions used for regional analyses. Stations are coloured by the starting month of their fire season, defined as the 4-month period with the highest average FWI. Coverage was best over the southern Canadian part of Boreal North America (BONA), Temperate North America (TENA), Europe (EURO), the Central Siberian part of Boreal Asia (BOAS), Japan and the southern China regions of Central Asia (CEAS) and coastal Australia (AUST). Coverage was reasonable over Central America except for north-Central Mexico, and the Malaysian and western Indonesia part of Equatorial Asia (EQAS). Coverage was otherwise poor, notably over the fire prone regions of South America (SAM) such as the Mato Grosso of Brazil, all of Africa (AFR), Southeast Asia (SEAS) except for Thailand, central Asia, and western Russia.

8-day FWI forecasts calculated from GEOS-5 weather forecasts were evaluated for 2018, the first full year for which forecasts have been produced. Forecasts were analysed over the same Terrestrial Ecoregions of the World boundaries (Olson et al., 2001) as the fire-climate analysis of Abatzoglou et al. (2018), rather than GFED regions, which were

judged to be too big, or state or provincial boundaries, which were judged to be too small. The forecasts at each lead time were compared to the GEOS-5 FWI analysis fields (i.e. 0-day lead time fields from the data assimilation system which are observationally constrained), rather than to FWI calculated from weather stations because of the low weather station density over many fire prone regions of the world.

**3 Results**

**3.1 MERRA2 and station FWI comparison over 2004-2018**

**3.1.2 Examples for Canada and Spain**

To illustrate differences between the station and MERRA2-based weather inputs and FWI System component values, two examples of daily data are provided for weather stations in different fire environments during which there were significant fire events, and in countries where the FWI System is used operationally.

Figure 2 shows the daily 12:00 local time 2m temperature (TEMP) and relative humidity (RH), 10m wind speed (WDSPD), 24-hour precipitation (PREC) and the individual FWI component values for Ft. McMurray, Alberta, in western Canada for 2016. The Ft. McMurray wildfire of May 2016 destroyed over 3000 structures in the city of Ft. McMurray and led to the largest evacuation in Canadian history. Station-based FWI calculations began in mid-April after the snow melt, which was followed by warming and drying conditions through end of the month. The fire was first detected on May 1[st] when the FWI was 28, which would be classified as Very High in Alberta (Stocks et al., 1989), and until May 8th varied between 40 to 46, which would be classified as Extreme. These conditions were driven by an absence of rain during the prior two weeks and low (< 30%) RH. The MERRA2-based FWI only marginally captured the extreme fire weather conditions, due primarily to a combination of too-late snow melt, too-high RH during May, and too-low windspeeds.

Figure 3 shows the daily weather and FWI values for Vigo in northwestern Spain over 2017. Beginning in April, the station-based DMC increased over the summer, punctuated by periodic decreases associated with small rain events. The DC increased more steadily, due to it being less sensitive to small amounts of rain. By October, BUI values exceeded 100, which would represent low fuel moisture content in heavy and medium-sized dead fuels, and a very dry landscape. The severe burning on October 15 was associated with FWI of 72 for the weather station data and 50 from MERRA2, which would be classified as Extreme in southern Europe (Palheiro et al., 2006;San-Miguel-Ayanz et al., 2013). The MERRA2-based calculations for Vigo captured the progression of seasonal fire weather much better than for Ft. McMurray.

**3.1.2 Global FWI means and biases**

Figure 4 shows the mean values for each of the six FWI components calculated from ISD stations with sufficiently complete data over 2004-2018, calculated only over the local 4-month fire season beginning on the month shown in Figure 1. The FFMC (Figure 4a) generally has a mean FFMC greater than 75, with higher values seen over the western

US, southern Europe, south-western Siberia, Thailand and most of Australia. Lower mean values are seen over the western and eastern Canadian coasts, the UK, northern Europe, southern China and the Maritime continent. The DMC (Figure 4b) mean values range from below 50 across most of Canada, the eastern US, north and central Europe, Siberia, China and the southeast coast of Australia, to above 300 over the western US and northern Australia. Patterns in the mean DC (Figure 4c) follow those of the DMC, but with a maximum of 1000 over the southwest US, southern Spain, and parts of Australia. The BUI (Figure 4e) has the same pattern as the DMC and DC, but over a range up to 350. The patterns of ISI (Figure 4d) and FWI (Figure 4f) follow those of the other indices, with maximum means of 25 and 60 respectively.

Figure 5 shows the bias of MERRA2 FWI components relative to the station data over the local 4-month fire season, and Figure 6 shows the bias of the input weather variables. The FFMC (Figure 5a) had a median bias of -0.2 over all stations. This was a mix of the coherent low biases over the most of Canada, central America, northern Eurasia, the western Maritime Continent, and coastal Australia, with weak positive biases over the Canadian Plains and central Europe. Qualitatively, the spatial patterns in FFMC reflect the biases in TEMP (Figure 6a) and RH (Figure 6b). The median DMC bias was -6.1 (Figure 5b) which reflected strong negative biases over the western north America and northern Australia, with no comparable regions of coherent high bias, and no clear relationship to the patterns in the individual input variables. The DC (Figure 5c) had a median bias of -54.7, with strong low biases over the western US and the Australian interior, coherent but weaker low biases over Canada and most of Eurasia, and a slight but coherent high bias over the southeast US and southwestern Australia. Like the DMC, there was no clear association between DC biases and either the TEMP or PREC biases, but the low biases over the western US were consistent with too much snow (Figure 6e) and a shorter period of active FWI calculations (Figure 6f). The ISI (Figure 5d) is mostly biased low, and most strongly over the western US. There are areas of high ISI bias in central Canada, Spain, central Europe, Thailand, and southwest and northern Australia. The patterns in ISI bias weakly reflect those of the WDSPD (Figure 6c). The bias pattern in BUI (Figure 5e) is nearly identical to that of the DMC, and that of the FWI (Figure 5f) to the ISI. Maps of relative rather than absolute biases tended to minimize the dominance of biases in regions with high mean FWI component values (e.g. the western US and northern Australia) make biases in either direction in other regions more apparent.

To quantify the relationship between MERRA2 FWI component biases and those of the input weather variables, Table 2 summarizes weather and FWI means for weather stations, MERRA2 biases, and the correlations between FWI component biases from Figure 5 and weather input biases from Figure 6. SNOWD is the percentage of days with snow on the ground and FIRESEASON is the percentage of days during which the FWI calculations are active. Globally, MERRA2 has a -0.3 °C temperature bias, a -0.6% RH bias, a -0.2 kph windspeed bias, a 0.6mm/day precipitation bias, 7.8% too many days with snow, and a 4.5%-day shorter fire season. To identify which individual weather bias might most influence FWI component biases, the interior values of Table 2 (in italics) show the correlations between biases in weather and biases in FWI components across stations. Biases in the FFMC are positively related to biases in TEMP (r=0.73) and negatively related to biases in RH (r=-0.72), and secondarily to PREC biases (r=-0.50), with little relation

(r=0.17) to WDSPD biases. Globally, biases in the DC, DMC and BUI are not strongly related to biases in any individual weather input. Biases in the ISI are moderately related (r=0.56) to biases in the windspeed, with a slight negative relationship (r=-0.47) with RH. Biases in the FWI component are most strongly related to TEMP, RH and WDSPD, through the intermediate biases of the FFMC and the ISI.

Globally-averaged FWI and biases obscure considerable regional variation in weather and FWI biases and relationships to biases in the weather inputs because of the large variation in mean FWI component values between, for example, arid and tropical fire environments. The same statistics shown in Table 3 were calculated across stations for each of the GFED regions. Table 3 shows the mean station weather and FWI values, MERRA2-biases and bias correlations for the BONA, TENA, CEAM, SAM regions. Over BONA, the relationships between FFMC biases and weather biases were consistent with the global relationships, but stronger (r=0.82 for TEMP, r=-0.81 for RH, r=-0.59 for PREC). Biases in the DMC and BUI were related to biases in TEMP, and the DC biases to biases in TEMP and PREC (r=-0.66). ISI biases were related to biases in TEMP and RH via the FFMC and to biases in WDSPD (r=0.67). Biases in the FWI were most strongly related to temperature biases (r=0.76) via the individual sub-components, and also to RH and WDSPD. It should be noted that the agricultural regions of the Canadian Prairies are overrepresented in these estimates, and the wildfire-prone areas of northern Canada under-represented. Over TENA, the FWI components were also biased low, reflecting strong biases in the west compared to the east. The weather bias influence on FWI component biases was generally weaker than BONA, aside from a strong influence of RH bias (r=-0.83) on the FFMC. Biases in the FWI were most strongly (r=0.63) related to TEMP. There was a weak (r=0.46) relationship between DC biases and FIRESEASON, suggesting that too late a start in the DC calculations led to less 'drought accumulation' over the fire season, particularly in the western US.

CEAM FFMC biases were most strongly related to PREC (r=-0.84) and RH (r=-0.68), which translated into strong bias relationship on the FWI for the RH (r=-0.82). The DMC, DC and BUI were biased low, but with only a weak (r=-0.52) influence from PREC biases, and no relationship to TEMP biases, due to less variation during the fire season. SAM biases were harder to quantify because of poor station coverage. Across the 21 stations that were available, there were strong low biases in all FWI components, which had similar relationships to weather input biases as CEAM. SNOWD and FIRESEASON were related to the DMC, DC and BUI, but this was due to a single outlying station at the Santiago airport in Chile (WMO ID 855740).

Table 4 shows mean and bias statistics for AFR, EURO, BOAS and CEAS. Like SAM, there were very few (n=10) stations over AFR. All FWI components were biased low, with FFMC and DC biases related to PREC biases, DMC, ISI and FWI most strongly related to RH biases, but with too few stations for these relationships to be considered robust. EURO had good station coverage spanning the different fire environments of the Mediterranean to Scandinavia. FWI component biases were negative, but lower in magnitude than globally. FFMC biases were strongly related to TEMP (r=0.78), RH (r=-0.80), PREC (r=-0.71), and weakly to FIRESEASON (r=0.55). There were moderate TEMP (r=0.51) and RH (r=-0.54) relationships with the DMC, and also between PREC biases and DC biases

(r=-0.66), and with a weak (r=0.43) relationship to FIRESEASON. FWI biases were more strongly related to RH biases (r=-0.70) than to TEMP (r=0.57) and PREC (r=-0.52) biases.

BOAS had low biases across all FWI components, but which were representative almost entirely of Siberia. FFMC biases were similarly related as EURO for biases in TEMP (r=0.75), RH (r=-0.86) and PREC (r=-0.74), and with no strong snow day or fire season length influence. TEMP, RH, PREC and FIRESEASON influences on the DMC, DC and BUI were comparable to EURO, and ISI biases had strong relationships with RH (r=-0.69) and WDSPD (r=0.63) biases. The strongest relationships with FWI biases were for RH biases (r=-0.71), PREC (r=-0.64) and TEMP (r=0.64). Stations over CEAS were primarily in southern China and Japan, and all FWI components were biased low except for the ISI. The FFMC biases were related to TEMP (r=0.76), RH (r=-0.72), with no strong relationships for DMC, DC or BUI biases, and moderate relationships for TEMP and RH for both the ISI and FWI.

Over SEAS (Table 5), FWI component biases were more weakly low compared to other regions, and slightly high for the ISI and FWI, but reflect coverage primarily over Thailand, with scattered stations in Vietnam, Myanmar and Pakistan, and with no coverage over India or Bangladesh. FFMC biases were related to RH (r=-0.79) and PREC (r=0.76) biases. FWI biases were most strongly related to RH (r=-0.76) and TEMP (r=0.73) biases. The strong negative relationships between the DMC and BUI with FIRESEASON were due to outlier values for four stations over Pakistan and are not likely robust. Indeed, when these stations were excluded from the analysis, the DMC and BUI correlations with FIRESEASON were reduced to r=-0.26 and r=-0.22 respectively. Over the tropical EQAS region, low mean FWI values reflect tropical conditions, for which MERRA2 FWI component values were further biased low. Biases relationships were generally weaker than over SEAS, with RH biases having the strongest relationships to FFMC (r=-0.63) and ISI (r=-0.62) biases, and DC biases being moderately (r=-0.59) related to PREC biases. Overall, the weak bias relationships reflect little spatial variation in the average FWI component values.

AUST had good station coverage, showing high average FWI values in the interior and west coast, and the lower average values along the other coasts, Tasmania and New Zealand. Mean biases in FWI components were negative except for the FFMC, but smaller in magnitude than other regions due to smaller biases in the weather inputs. FFMC biases were strong related to TEMP (r=0.90) and RH (r=-0.82) biases. The strongest relationships with the FWI were with biases in TEMP (r=0.60) and RH (r=-0.59).

### 3.2.2 Global FWI temporal correlations

To understand the degree to which MERRA2 FWI components capture daily changes in station FWI, Figure 7 shows the correlation at each station between the daily station and MERRA2 FWI component values during each station's 4-month fire season. The histogram inset in each panel shows the frequency distribution of the correlations across stations. The histograms also show the frequency distribution of correlations calculated using 3, 7 and 30-day averages of the daily time series, which reflects the time scales over which fire weather analyses are done. Figure 8 is similar, but for the input weather variables.

The MERRA2 and station FFMC (Figure 7a) are correlated at stations over northern midlatitudes (TENA and EUR), weakening somewhat over BONA and BOAS, and correlations are lower over the tropics, most clearly seen over Thailand, Malaysia and the Philippines. The median correlation across all stations increases from r=0.75 for daily FFMC to r=0.79 for 3-day averages, r=0.81 for 7-day averages, and r=0.83 for 30-day averages, and the frequency distribution becomes more left-skewed for longer averaging windows. Globally, the spatial correlation distribution most closely follows that of the correlation between MERRA2 and station RH (Figure 8b), as does the change in frequency distribution with averaging period, though with a progressively flatter peak for the RH.

Correlations between daily station and MERRA2 DMC (Figure 7b) are lower than for the FFMC, with a median correlation of r=0.68 for the daily time series. Areas of low correlation for the DMC are over the central US, northern Canada, south-central Siberia, central China, Thailand and Malaysia. The DMC correlations are less sensitive to the averaging period than the FFMC but increases to r=0.73 for a 30-day average. The DMC correlation pattern corresponds to that of the PREC correlation (Figure 8d). For different averaging periods, the change in frequency distributions of DMC correlations appears to be limited by that of the change in PREC correlations.

DC correlations are higher than for the DMC (Figure 7c) for the daily time series, and are less strongly related spatially to those of PREC because of less sensitivity to individual precipitation events. Longer averaging periods have no effect on DC correlations because the DC is less sensitive to how the precipitation is distributed over time. ISI correlations (Figure 7d) are most closely related to WDSPD correlation patterns (Figure 8c), seen most clearly over North America and Australia. The change in frequency distribution of ISI correlations reflects those of the FFMC and WDSPD. The BUI (Figure 7e) correlation patterns for daily data follow those of the DMC, but are higher due to the influence of the DC. The FWI (Figure 7f) correlation patterns follow those of the ISI, as does the rightward shift in the frequency distribution of correlations with increasing averaging period.

### 3.2 GEOS-5 FWI forecast evaluation for 2018

### 3.2.1 Example for the 2018 fire season over central British Columbia, Canada

The forecast evaluation focuses on the FWI component. To illustrate the performance of FWI forecasts over a single region, we use the severe 2018 fire season over west-central British Columbia (BC), Canada (Tollefson, 2018). The FWI System is used operationally in BC, with prevention and pre-preparedness measures tied to joint BUI/FWI thresholds (Stocks et al., 1989). For simplicity, we interpret the 2018 FWI variation using the 'marginal' FWI thresholds. Figure 9 shows the 137 065 km$^2$ Fraser Plateau and Basin Complex ecoregion from the Terrestrial Ecoregions of the World, where several of the largest fires burned. This corresponds roughly to the BC government's Interior Plateau Region II, where an FWI of greater than 31 is considered extreme (Stocks et al., 1989). For context, the 500 hPa heights for the first three weeks of August 2018 leading up to the peak in fire activity are also shown. The relevant feature is a persistent ridge of high pressure extending from the southwest US to the Yukon, which is

associated with warm and dry conditions in BC, and, historically, higher fire activity in western Canada (Skinner et al., 1999).

Figure 10a shows the daily MODIS active fire counts and GEOS-5 analysis FWI averaged over the Fraser Plateau and Basin Complex ecoregion. The FWI System calculations start up at the end of April after snow melt, and the FWI remains below 10 through May and June. The FWI increases over July and early August, and is punctuated by two rain events from which the FWI recovered after several days. Under warm and dry conditions, the FWI mostly

remained above 20 for the first three weeks of August, during which several large fire complexes grew, shown by the increase in daily MODIS active fires, which peaked on August $22^{nd}$.

Figure 10b shows the forecast FWI over the region at lead times of 1 to 8 days using the approach of Carbin et al. (2016). The FWI colour scale is similar to that of the Global Wildfire Information System

(http://gwis.jrc.ec.europa.eu), which reflects a wider range than that over BC. The shaded FWI on the bottom row of the panel with lead-time 0 corresponds to the FWI time series in the top panel, and represents the forecast target on different days. Reading upward, each row shows the forecast with increasing lead time; a perfect forecast over lead times of up to 8 days would be shown by a vertical line with the same colour as that on the target date.

For May and June, the forecasts capture the low FWI for lead times of up to 8 days. The observed increase in FWI mid-July is captured at lead times of up to 5 days, as is that at the end of the month. The low FWI of 10 at the beginning of August following a 1-day rain event is captured by the forecast up to 8 days in advance. At the end of the first week of August, there was lower FWI forecast between 4 and 5 days in advance, indicated by the isolated patch of blue, and which did not strongly verify. The forecasts captured the increase toward high (>20) FWI in mid-August, and the peak

FWI of 29 on August 22, when fire activity was at its highest. This was followed by lower FWI for September and October which was well-forecast, including several brief FWI calculation 'shutdowns' before the final shutdown at the end of October. During the May-October fire season, the correlation between the daily analysis and forecast FWI was r=0.96 at 2-days lead time, r=0.88 at 4-days lead time, r=0.82 at 6-days lead time, and r=0.68 at 8-days lead time.

### 3.2.2 Global FWI forecast correlations and biases

The maps in Figure 11 show the correlation between analysis and forecast FWI at lead times of 1 to 8 days, for FWI values averaged over each of the 771 Terrestrial Ecoregions of the World, excluding unvegetated areas. For each ecoregion, only the four consecutive months with the highest mean FWI were considered. As with the comparison of station and MERRA2 FWI, this was done to reduce the influence of wet and dry seasonality in the tropics in the correlations, and to make for a more meaningful forecast comparison between regions with year-round versus partial-

355    year fire seasons.

At a lead time of 1 day, there is mostly perfect correlation between the forecast and analysis FWI across all ecoregions, with slightly lower values in the eastern US, southern South America, the Sahel, southern Africa, and South Asia. At

a lead time of 3 days, correlations are less than 0.80 over parts of northern Canada, the southeast US, northern Africa and South Asia, but otherwise remain high. At a lead time of 5 days, there is a broad arc of low (r<0.50) correlation stretching across northern Canada, and lower correlations over the eastern US. Correlations also decrease over southern South America, southern Africa and northern Africa adjacent to the Sahara, Siberia, South Asia and the ecoregions in SEAS and EQAS along the Pacific Rim. At lead times of 7 and 8 days, there is a wide range of correlations between forecast and analysis FWI. Correlations are high (r > 0.80) over parts of the western US, central America and northern South America, central Africa, parts of the Mediterranean, and southern China and northern Southeast Asia, but are otherwise very low.

Figure 12 shows the distribution of ecoregion correlations between forecast and analysis FWI at different lead times, organized by the GFED regions. The decay of forecast skill is captured by how much the distribution shifts leftward with increasing lead time. Over Boreal North America (BONA) and Boreal Asia (BOAS), there is a steady leftward shift in the distribution, and flattening of the distribution after a lead time of 4 days. The faster decay in forecast skill over cold regions is in part due to less variability in FWI and larger ecoregions with more within-region variation in FWI. Over Temperate North America (TENA) and Australia (AUST), there is a slower leftward shift in the distribution and sharper peaks around median correlations of 0.58 and 0.49, respectively. Over Central America (CEAM), South America (SAM) and Africa (AFR), by contrast, the forecast skill deteriorates more slowly, with median correlations of greater than 0.80 at lead times of 5 days.

Figure 13 shows the bias between forecast and analysis FWI for each ecoregion. At high northern latitudes, there is no discernible systematic bias in the FWI forecasts for any lead time, but this is in part a function of the narrower FWI scale over those regions. Moving equatorward, the FWI forecasts are in general biased high, which is most apparent over the US, south-eastern Brazil, and South Asia. This bias increases with lead time, but is less apparent than the decay in correlation with lead time in Figure 12. Compared to the decay in correlation, the biases do not increase as significantly with lead time. Figure 14 shows the distribution of forecast biases across ecoregions with increasing lead time, organized by GFED region. Over BONA and BOAS, there is no systematic change in bias with lead time, but across all other ecoregions, the distribution of biases shifts rightward with increasing lead time.

The skill of the FWI forecasts will depend on the forecast skill for the underlying weather input values. There was no association between the regional differences in FWI correlation decay with lead time in Figure 11 and those for TEMP correlation, which decreased more slowly (not shown). The decrease in PREC forecast correlation is shown in Figure 15. There is some association between patterns in decrease in FWI forecast correlation and precipitation correlation, but the latter tends to decrease more quickly with increasing lead time. Over North America, for example, the north-eastward decrease in FWI skill is only weakly apparent in the precipitation map. There was a stronger association with RH forecast correlation, shown in Figure 16. For lead times of greater than 4 days, there is a more apparent relationship between patterns of FWI and RH forecast skill at continental scales.

## 4 Discussion

For the FWI fields calculated from MERRA2 weather inputs, the dependence of biases in the FWI components on weather inputs varied by component and region. Of any single input, biases in the TEMP and RH across stations tended to be correlated with biases in the FWI components most frequently across GFED regions. Systematic, persistent biases in the TEMP and RH will continually affect the moisture codes, whereas PREC, even if biased, is more episodic, and will also be buffered slightly by the precipitation thresholds for the wetting phases of the moisture codes, and in the FFMC, a fast recovery from individual precipitation events. Relationships between WDSPD biases and ISI biases were present in several regions (AUST, BOAS, CEAM, BONA), but with weaker relationships to FWI biases because of the influence of other inputs and intermediate FWI components. These biases should be taken account when using MERRA2 based FWI for fire-climate analyses, and should be the focus, alongside precipitation, of bias-correction efforts in computing fire weather indices from analysis and forecast fields. Bias in the FIRESEASON lengths were related to biases in the DC over northern mid-latitudes, presumably because of less drying time over which the DC can increase during the fire season.

The biases seen in MERRA2-based FWI were generally consistent with comparisons to station for other fire weather products, at least in sign. In comparing station to high-resolution analysis MacArthur Forest Fire Danger Index over south-eastern Australia, Clarke et al. (2013) found a change from positive to negative analysis field bias moving from the interior to the coast. Over that region, the FWI (Figure 5f) shows no positive bias inland, but the negative bias does strengthen toward the coast, due primarily to corresponding gradient toward stronger low wind-speed biases in MERRA2. Over the Great Lakes region, Horel et al. (2014) found that the FWI components calculated from high-resolution analysis fields were biased low except for the DMC, which was consistent with the biases in Figure 5. Over Spain, there was a change in FWI bias from high to low moving toward the Mediterranean coast (Figure 5f), which was also seen in Bedia et al. (2012) for 7 stations, particularly for FWI calculated from the National Centers for Environmental Prediction / National Center for Atmospheric Research reanalysis.

There are no global evaluations of short-term fire weather forecast skill against which the FWI forecasts can be compared, but several comparable regional studies have been conducted. Using high resolution WRF forecasts, Mölders (2010) found that FWI forecasts June of 2005 in the interior of Alaska were skilful, with little decrease in skill for leads of up to 5 days. The GEOS-5 based FWI forecasts showed a slight decrease over the ecoregions of southern Alaska, but also remained skilful at lead times of up to 5 days, presumably because of the ability of the GEOS-5 model to resolve large-scale weather systems arriving from the Pacific, but there was a significant drop in skill in terms of forecast-analysis correlations over this region for leads of 6-8 days, however. Horel et al. (2014) found that FWI forecasts over the US Great Lakes Region for the 2012 fire season, bias and RMSE of the forecasts relative to station data did not increase significantly for leads of 24- and 48-hours, consistent with the GEOS-5 based FWI forecasts over that region, which, compared to Alaska, remained skilful at longer lead times. Freitas et al. (2018) compared the GEOS-5 500 hPa height global anomaly pattern correlations for lead times of up to 5 days. For either convective parameterization considered, there was a pronounced decrease in skill for forecast leads of 3-5 days

compared to 1-2 days. To the extent that the local fire weather is controlled by the large-scale circulation, this is likely reflected in a similar drop in skill for many regions in Figure 11 and Figure 12 beyond lead times of 2 days, particularly in the extratropics. Although at a seasonal time-scale, Bedia et al. (2018) found that seasonal FWI predictions over Europe using the ECMWF System 4 seasonal climate forecasts were influenced by the skill of relative humidity predictions, consistent with its importance over short forecasts examined here.

## 4 Conclusions

Meteorological analyses provide the only practical means of making fire danger products at global scales, but these should be accompanied by estimates of these products' biases relative to weather station data. This study has done so for the MERRA2 reanalysis and identified the contributions of biases in different input weather variables to biases in FWI System components at continental scales. The focus of earlier MERRA and MERRA2-based evaluation was precipitation from rain gauges (Field et al., 2015) and satellites (Field, in press); this study has shown that biases in temperature and relative humidity also need to be considered. While important, errors in reanalysis precipitation affect FWI values episodically, whereas persistent errors in temperature and relative humidity affect the FWI values continuously.

Considering these discrepancies is particularly important for any practical application of the data, which inevitably require fire-environment specific interpretation. The studies listed in Table 1are representative of FWI interpretation in different fire environments and are mostly based on FWI components calculated from weather station data. The fire danger classifications therein will not necessarily be applicable to FWI values calculated from reanalysis or analysis fields. Two alternatives are to re-develop fire danger classifications for the particular data product (as examined by Vitolo et al. (2018)) or to apply bias corrections to the input weather data values or calculated FWI values, using, for example techniques applied to climate model projections which correct for biases in the models' biases for present-day climate (Yong et al., 2015;Casanueva et al., 2018). The latter approach will require enough high quality, hourly weather station data, which was found to be limited for Northern Canada, South America, Africa, the Middle East, central Eurasia and South Asia, similar in this study and also in Vitolo et al. (2019). In that regard, future evaluation of these products would benefit from high-quality hourly weather data archived national meteorological agencies, and state/provincial level agencies operating secondary weather networks. Data from secondary station networks which is not assimilated into reanalysis products will also be useful for providing a fully independent evaluation. The 2004-2018 period of this study was constrained by the availability of MODIS snow cover data needed to supplement station records. Longer-term station records would be helpful in determining how changes in reanalysis input data sources, particularly for infrared and microwave radiances (as described in Gelaro et al. (2017)) translate into changes in surface temperature, humidity and precipitation fields.

This study also provided a first, if limited, evaluation of global FWI forecast skill. For 2018, forecasts at lead times of 1-2 days were very highly correlated with the analysis FWI, and at longer lead times, correlations decreased more at high latitudes. Forecasts at lead times of 7-8 days were largely unskilful, and more spatially incoherent, which serves

as a reminder that despite their availability, longer-lead fire weather forecasts from global models have very limited utility. It will be important in future work to see if skill for different years is comparable, and also for more individual

fire events. As the use of fire weather from global analysis and forecast fields becomes more widely used, systematic comparisons of different models will also be useful.

## Code and data availability

All code and data are accessible through the GFWED website: https://data.giss.nasa.gov/impacts/gfwed/.

## Acknowledgements

This work was supported by the NASA Group on Earth Observations Work Program grant number 80NSSC18K0410.

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

**Tables**

**Table 1. Examples of FWI calibration and adaptation studies for different fire environments around the world.**

| Region | Weather data | Indices | Approach | Reference |
|---|---|---|---|---|
| Alberta, Canada | On-site measurements | FWI | Experimental fire behavior examples for different FWI values in a reference Jack Pine fuel type in relation to fire intensity and suppression difficulty. | Alexander and de Groot (1988) |
| Canadian provinces | Weather stations | FWI, FWI & BUI for BC | Cumulative FWI frequency distributions, relationships between FWI, number of fires and burned area from reports, expert assessment. | Stocks et al. (1989) |
| Northeast China | Weather stations | All | Cumulative FWI frequency distributions, number of fires and burnt area across fire danger classes, relationships between FWI values, number of fires and area burnt. | Tian et al. (2011) |
| Southwest Slovenia | Weather stations | FWI | Cumulative FWI frequency distributions, number of fires and burnt area across fire danger classes, logistic regression between FWI indices and days with fire. | Sturm et al. (2012) |
| Districts in Portugal | Weather stations | FFMC, DC, ISI, FWI | Cumulative FWI frequency distributions, relationships between FFMC and moisture content of dead Eucalyptus leaves, ISI and spread rate in shrub vegetation, DC and live moisture content of shrubs, DC and total annual June-Sept area burned. | Fujioka et al. (2009), translated from Viegas et al. (2004) |
| Portugal | Weather stations | FWI | Estimated fireline intensity and difficulty of suppression for maritime pine stands in Portugal using experimental fires and wildfires, simulated fire spread rates. | Palheiro et al. (2006) |

| | | | | |
|---|---|---|---|---|
| Crete, Greece | Single weather station | FFMC, DMC, FWI | Cumulative FWI frequency distributions, sub-index correlations with number of fires and burned areas from fire reports, relationships between FFMC and sampled fine fuel moisture content, DMC and sampled duff moisture content. | Dimitrakopoulos et al. (2011) |
| Patagonia, Argentina | Weather stations | FFMC | Relationships between FFMC and laboratory ignitions, and moisture content for cypress and shrub litter. | Bianchi and Defosse (2014) |
| United Kingdom | Weather stations and NWP analysis | All | Cumulative FWI frequency distributions relative to fire occurrence, emphasizing percentile-based classification, possible utility of absolute FFMC values. | de Jong et al. (2016) |
| Indonesia and Malaysia | Weather stations | FFMC, DC, ISI | Grass fuel ignition tests, satellite active fires, airport visibility as an indicator of severe haze. | de Groot et al. (2007) |
| General | ERA-Interim reanalysis | FWI | General fire weather index calibration software, regional European examples provided for satellite-based burned area. | Vitolo et al. (2018) |

**Table 2. Weather input and FWI statistics for 1746 weather stations and MERRA2 reanalysis fields sampled at station locations for 2004-2018. The first row in the table is the mean for each weather input from the weather station data, and the second row is the mean MERRA2 bias relative to the station data. The first column is the mean FWI value across weather stations, and the second column is the mean MERRA2 bias relative to the station data. The interior table entries in italics are the correlations (for $p < 0.05$ only) between the FWI component biases and the weather input biases across stations. Means and biases at each station are calculated only over the local 4-month fire season. SNOWD is the percentage of days with snow on the ground and FIRESEASON is the percentage of days during which the FWI calculations are active.**

| | | | | TEMP (ºC) | RH (%) | WDSPD (kph) | PREC (mm/d) | SNOWD (%) | FIRESEASON (%) |
|---|---|---|---|---|---|---|---|---|---|
| **Global** | n = 1746 | | **STN MEAN** | 23.6 | 52 | 14.3 | 2.3 | 17 | 76.8 |
| | | **STN MEAN** | **MERRA2 bias** | -0.3 | -0.6 | -0.7 | 0.5 | 7.8 | -4.5 |
| | **FFMC** | 80.3 | -1.3 | *0.73* | *-0.72* | *0.17* | *-0.50* | | *0.20* |
| | **DMC** | 67.3 | -12.7 | *0.39* | *-0.32* | | *-0.14* | | *0.15* |
| | **DC** | 353 | -64.3 | *0.30* | *-0.12* | *0.05* | *-0.37* | *-0.13* | *0.25* |
| | **ISI** | 8 | -0.8 | *0.41* | *-0.47* | *0.56* | *-0.12* | *-0.11* | *0.17* |
| | **BUI** | 83.7 | -15.3 | *0.41* | *-0.30* | *0.05* | *-0.20* | | *0.18* |
| | **FWI** | 19.7 | -2.2 | *0.57* | *-0.60* | *0.46* | *-0.24* | *-0.13* | *0.23* |

**Table 3. Same as Table 2, but for the BONA, TENA, CEAM and SAM regions.**

| | | | TEMP (ºC) | RH (%) | WDSPD (kph) | PREC (mm/d) | SNOWD (%) | FIRESEASON (%) |
|------|------|------------|------|------|------|------|------|------|
| BONA | n = 267 | STN MEAN | 19.3 | 56.4 | 13.7 | 2.2 | 41.2 | 50.8 |
| | STN MEAN | MERRA2 bias | -0.6 | -0.3 | -0.9 | 0.9 | 11.4 | -6.7 |
| | FFMC | 75.7 | *0.82* | *-0.81* | *0.39* | *-0.59* | *-0.15* | *0.29* |
| | DMC | 34.9 | *0.68* | *-0.56* | *0.23* | *-0.41* | | *0.26* |
| | DC | 253.5 | *0.57* | *-0.37* | *0.25* | *-0.66* | *-0.17* | *0.34* |
| | ISI | 5.2 | *0.66* | *-0.65* | *0.67* | *-0.27* | *-0.16* | *0.32* |
| | BUI | 47.3 | *0.68* | *-0.54* | *0.23* | *-0.44* | | *0.30* |
| | FWI | 12 | *0.76* | *-0.70* | *0.53* | *-0.35* | *-0.15* | *0.35* |
| | | | | | | | | |
| TENA | n = 401 | STN MEAN | 27.1 | 47.5 | 15.3 | 2.6 | 9.4 | 81.8 |
| | STN MEAN | MERRA2 bias | -0.4 | 0.4 | -2.5 | 0.1 | 13.4 | -8.7 |
| | FFMC | 83.3 | *0.74* | *-0.83* | *0.33* | *-0.23* | | |
| | DMC | 85 | *0.53* | *-0.25* | *0.12* | *-0.22* | | *0.25* |
| | DC | 364 | *0.41* | | | *-0.47* | *-0.32* | *0.46* |
| | ISI | 10.3 | *0.36* | *-0.30* | *0.50* | *-0.14* | | *0.20* |
| | BUI | 100.5 | *0.53* | *-0.22* | *0.12* | *-0.27* | *-0.10* | *0.29* |
| | FWI | 24 | *0.63* | *-0.50* | *0.51* | *-0.31* | *-0.21* | *0.35* |
| | | | | | | | | |
| CEAM | n = 43 | STN MEAN | 28.7 | 50.5 | 13.6 | 1.2 | 0 | 99.2 |
| | STN MEAN | MERRA2 bias | 0 | -8.2 | -0.5 | 1.2 | 0.3 | -0.1 |
| | FFMC | 86.6 | *0.44* | *-0.68* | *0.42* | *-0.84* | | |
| | DMC | 154.1 | *0.47* | *-0.52* | | | | |
| | DC | 659.9 | | | | *-0.52* | | |
| | ISI | 10.1 | *0.48* | *-0.82* | *0.57* | *-0.33* | | |
| | BUI | 183 | *0.41* | *-0.47* | | *-0.35* | | |
| | FWI | 30.2 | *0.52* | *-0.82* | *0.50* | *-0.48* | | |
| | | | | | | | | |
| SAM | n = 21 | STN MEAN | 23.3 | 59.2 | 16.8 | 2.9 | 4.7 | 93 |
| | STN MEAN | MERRA2 bias | -0.2 | -2.2 | -2.4 | 2.4 | 3.8 | -3.7 |
| | FFMC | 79.4 | *0.52* | *-0.58* | | *-0.63* | | |
| | DMC | 38.5 | | | | | *-0.63* | *0.72* |
| | DC | 251 | | *0.64* | | | | *0.45* |
| | ISI | 6.9 | *0.45* | *-0.64* | *0.58* | | | |
| | BUI | 51.4 | | | | | *-0.70* | *0.80* |
| | FWI | 14.6 | *0.54* | *-0.60* | *0.45* | | | |

**Table 4. Same as Table 2, but for the AFR, EURO, BOAS and CEAS regions.**

| | | | | TEMP (ºC) | RH (%) | WDSPD (kph) | PREC (mm/d) | SNOWD (%) | FIRESEASON (%) |
|---|---|---|---|---|---|---|---|---|---|
| **AFR** | n = 10 | | STN MEAN | 28.6 | 67 | 10.2 | 2.4 | 0 | 98.8 |
| | | STN MEAN | MERRA2 bias | -1.5 | -0.1 | -1.2 | 1.1 | 0 | 0.7 |
| | FFMC | 79.7 | -6.8 | | | | *-0.88* | | |
| | DMC | 40.3 | -18.4 | | *-0.67* | | | | |
| | DC | 329.3 | -93.5 | | | | *-0.91* | | |
| | ISI | 3.8 | -1.6 | *0.77* | *-0.82* | *0.72* | | | |
| | BUI | 57.5 | -25.7 | | | | | | |
| | FWI | 10.6 | -5.3 | *0.63* | *-0.79* | | | | |
| | | | | | | | | | |
| **EURO** | n = 228 | | STN MEAN | 21.4 | 57.6 | 14.6 | 1.9 | 17.4 | 74.5 |
| | | STN MEAN | MERRA2 bias | 0 | -2.1 | -0.2 | 0.3 | 7.9 | -4.5 |
| | FFMC | 77.9 | -0.5 | *0.78* | *-0.80* | | *-0.71* | *-0.40* | *0.55* |
| | DMC | 58.5 | -3.8 | *0.51* | *-0.54* | | *-0.33* | | *0.19* |
| | DC | 357.8 | -37.8 | *0.53* | *-0.43* | | *-0.66* | *-0.28* | *0.43* |
| | ISI | 5.6 | -0.2 | *0.45* | *-0.58* | *0.44* | *-0.45* | | *0.16* |
| | BUI | 75.1 | -5.5 | *0.55* | *-0.55* | | *-0.42* | | *0.26* |
| | FWI | 15.3 | -0.4 | *0.57* | *-0.70* | *0.24* | *-0.52* | *-0.15* | *0.27* |
| | | | | | | | | | |
| **BOAS** | n = 161 | | STN MEAN | 18.6 | 56.8 | 10.3 | 2.3 | 51.3 | 42.6 |
| | | STN MEAN | MERRA2 bias | -0.4 | 0.7 | 0.8 | 0.4 | 8.1 | -4.7 |
| | FFMC | 74 | -2 | *0.75* | *-0.86* | *0.36* | *-0.74* | *0.29* | *-0.16* |
| | DMC | 26.9 | -5.8 | *0.70* | *-0.71* | | *-0.65* | | |
| | DC | 217.9 | -47.4 | *0.54* | *-0.27* | *0.17* | *-0.72* | *-0.38* | *0.46* |
| | ISI | 3.7 | -0.3 | *0.57* | *-0.69* | *0.63* | *-0.58* | *0.27* | |
| | BUI | 36.7 | -7.7 | *0.71* | *-0.67* | *0.16* | *-0.70* | | |
| | FWI | 8.4 | -1.3 | *0.64* | *-0.71* | *0.52* | *-0.64* | | |
| | | | | | | | | | |
| **CEAS** | n = 169 | | STN MEAN | 23.1 | 56.2 | 11.1 | 3.6 | 14.8 | 77.2 |
| | | STN MEAN | MERRA2 bias | -0.4 | 0 | 2.7 | 1.1 | 11 | -7.1 |
| | FFMC | 76.2 | -1 | *0.76* | *-0.72* | *0.19* | *-0.53* | | *0.25* |
| | DMC | 29.5 | -8.5 | *0.44* | *-0.23* | | *-0.18* | | |
| | DC | 179.1 | -61.5 | | *0.26* | | *-0.39* | *-0.32* | *0.37* |
| | ISI | 4.8 | 0.1 | *0.56* | *-0.66* | *0.49* | *-0.27* | | |
| | BUI | 38.3 | -11.5 | *0.39* | | | *-0.24* | | *0.19* |
| | FWI | 10.4 | -1.6 | *0.64* | *-0.62* | *0.32* | *-0.32* | | |

**Table 5. Same as Table 2, but for the SEAS, EQAS, and AUST regions.**

| | | | | TEMP (ºC) | RH (%) | WDSPD (kph) | PREC (mm/d) | SNOWD (%) | FIRESEASON (%) |
|---|---|---|---|---|---|---|---|---|---|
| SEAS | n = 63 | | STN MEAN | 29.4 | 56.5 | 8.2 | 2 | 0 | 99 |
| | | STN MEAN | MERRA2 bias | -0.4 | -4.9 | 3.2 | 0.5 | 0.1 | 0.4 |
| | FFMC | 83.8 | -0.3 | *0.67* | *-0.79* | *0.29* | *-0.76* | | *-0.35* |
| | DMC | 77.7 | -1 | *0.60* | *-0.54* | | *-0.64* | *0.35* | *-0.75* |
| | DC | 372.6 | -12.7 | *0.53* | *-0.42* | | *-0.63* | | *-0.57* |
| | ISI | 5.3 | 1.6 | *0.67* | *-0.71* | *0.55* | *-0.49* | | |
| | BUI | 95.7 | -1.9 | *0.61* | *-0.53* | | *-0.65* | *0.28* | *-0.72* |
| | FWI | 17.2 | 2.8 | *0.73* | *-0.76* | *0.36* | *-0.69* | | *-0.38* |
| | | | | | | | | | |
| EQAS | n = 39 | | STN MEAN | 30.2 | 68.5 | 9.6 | 6.5 | 0 | 99.1 |
| | | STN MEAN | MERRA2 bias | -2.3 | 6.8 | -1.5 | 2.1 | 0 | 0.3 |
| | FFMC | 72.8 | -14.8 | *0.62* | *-0.63* | *0.59* | | | |
| | DMC | 14.8 | -9 | *0.39* | *-0.39* | | | | |
| | DC | 110.9 | -36.6 | *0.33* | | | *-0.59* | | |
| | ISI | 2.8 | -1.7 | *0.51* | *-0.62* | *0.39* | | | |
| | BUI | 20.6 | -11.7 | *0.39* | *-0.32* | | *-0.37* | | |
| | FWI | 5 | -3.5 | *0.37* | *-0.43* | | | | |
| | | | | | | | | | |
| AUST | n = 344 | | STN MEAN | 24.4 | 42.6 | 18.7 | 1.2 | 0.2 | 97.4 |
| | | STN MEAN | MERRA2 bias | 0.3 | -1.2 | -1.6 | 0.3 | 0.5 | 0.9 |
| | FFMC | 86.3 | 0.1 | *0.90* | *-0.82* | | *-0.50* | | *0.68* |
| | DMC | 110.8 | -18.2 | *0.44* | *-0.52* | *-0.32* | | *0.15* | *0.18* |
| | DC | 554.4 | -87.5 | *0.28* | *-0.22* | | *-0.59* | | *0.33* |
| | ISI | 13.8 | -0.8 | *0.46* | *-0.45* | *0.64* | *-0.23* | | |
| | BUI | 137.5 | -21.5 | *0.47* | *-0.52* | *-0.28* | *-0.22* | *0.14* | *0.24* |
| | FWI | 34.9 | -2.6 | *0.60* | *-0.59* | *0.50* | *-0.37* | | *0.19* |

**Figures**

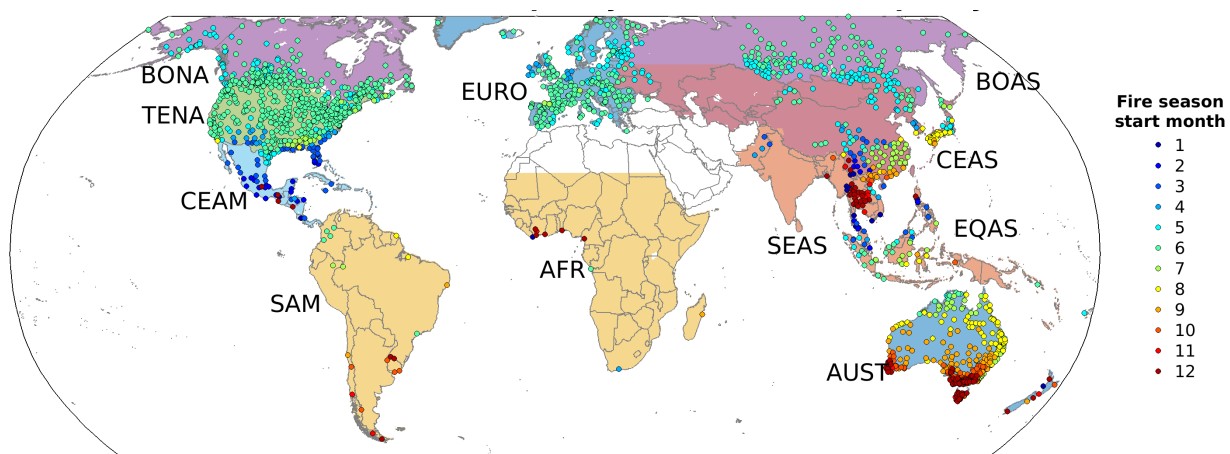

**Figure 1. National Centers for Environmental Information (NCEI) Integrated Surface Database (ISD) stations with at least 80% completeness of 12:00 local time observations of 2m temperature (TEMP) and 2m relative humidity (RH), and 50% completeness of daily total precipitation (PREC) over 2004-2018. Stations are coloured by the starting month of their 4-month peak fire weather season. Global Fire Emissions Database (GFED, van der Werf et al., 2017) regions are indicated by the labels and shading. The region definitions are: Boreal North America (BONA), Temperate North America (TENA), Central America (CEAM), South America, combining GFED Northern and Southern Hemisphere South America (SAM), Africa, combining GFED Northern and Southern Hemisphere Africa (AFR), Europe (EURO), Boreal Asia (BOAS), Central Asia (CEAS), Southeast Asia (SEAS), Equatorial Asia (EQAS), Australia and New Zealand (AUST). The GFED Middle East region was excluded due to a lack of weather stations.**

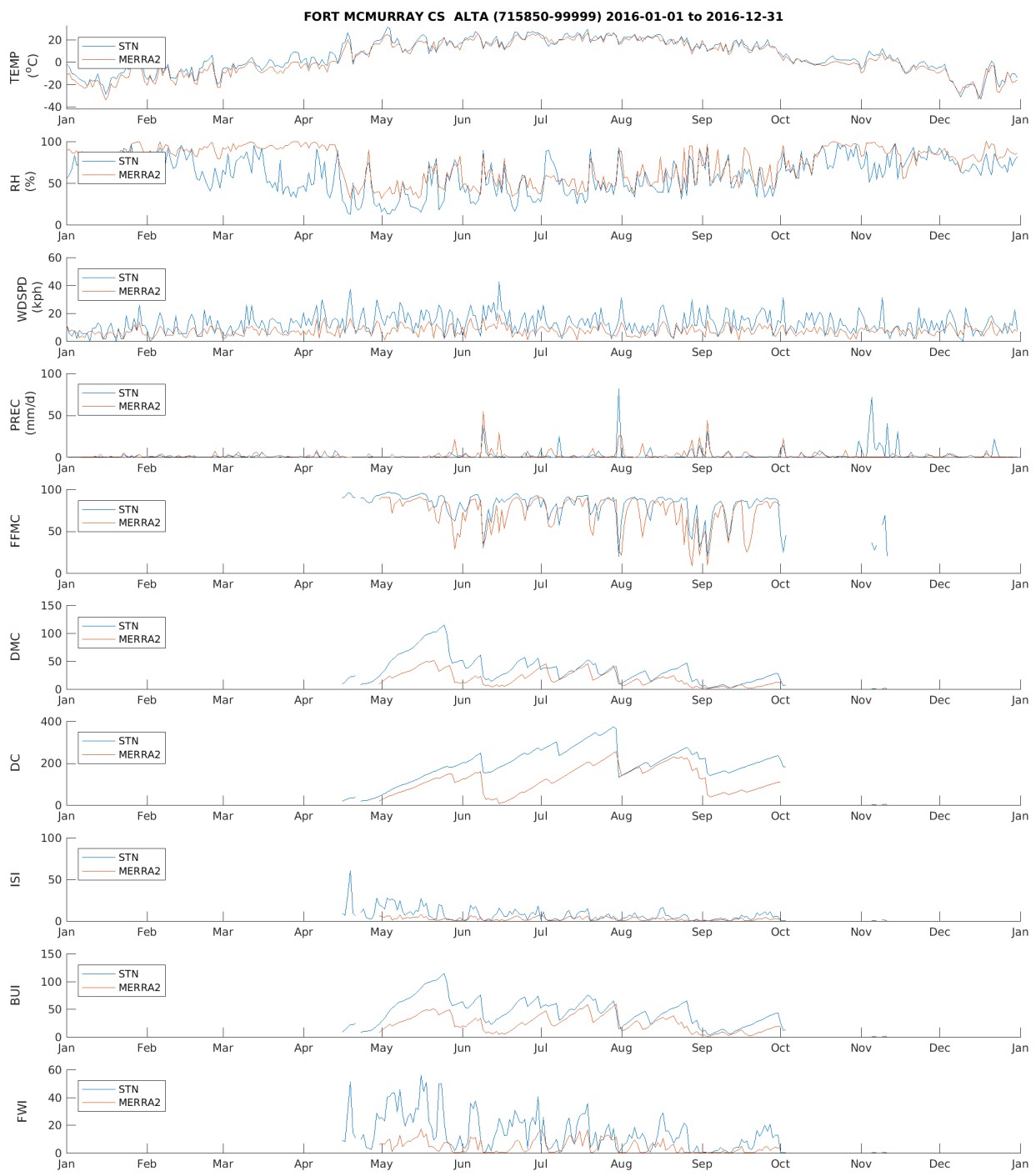

**Figure 2. Daily weather input and FWI System component values for Ft. McMurray, Alberta, Canada (WMO ID 715850, WBAN 99999, 56.65N, 111.22W) for 2016.**

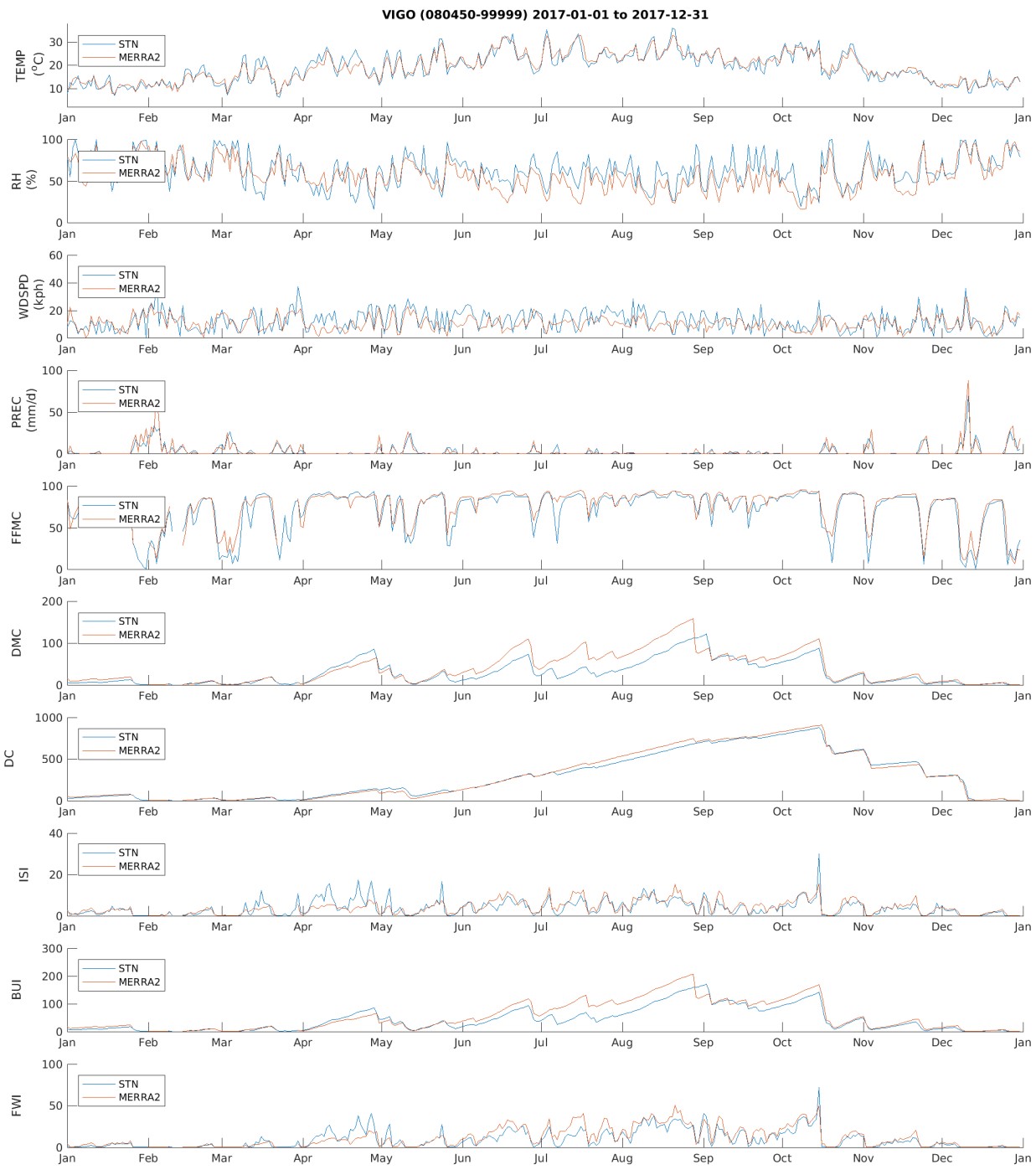

**Figure 3. Same as Figure 2, but for Vigo in northwestern Spain (WMO ID 080450, WBAN 99999, 42.232N 8.627W) during 2017.**

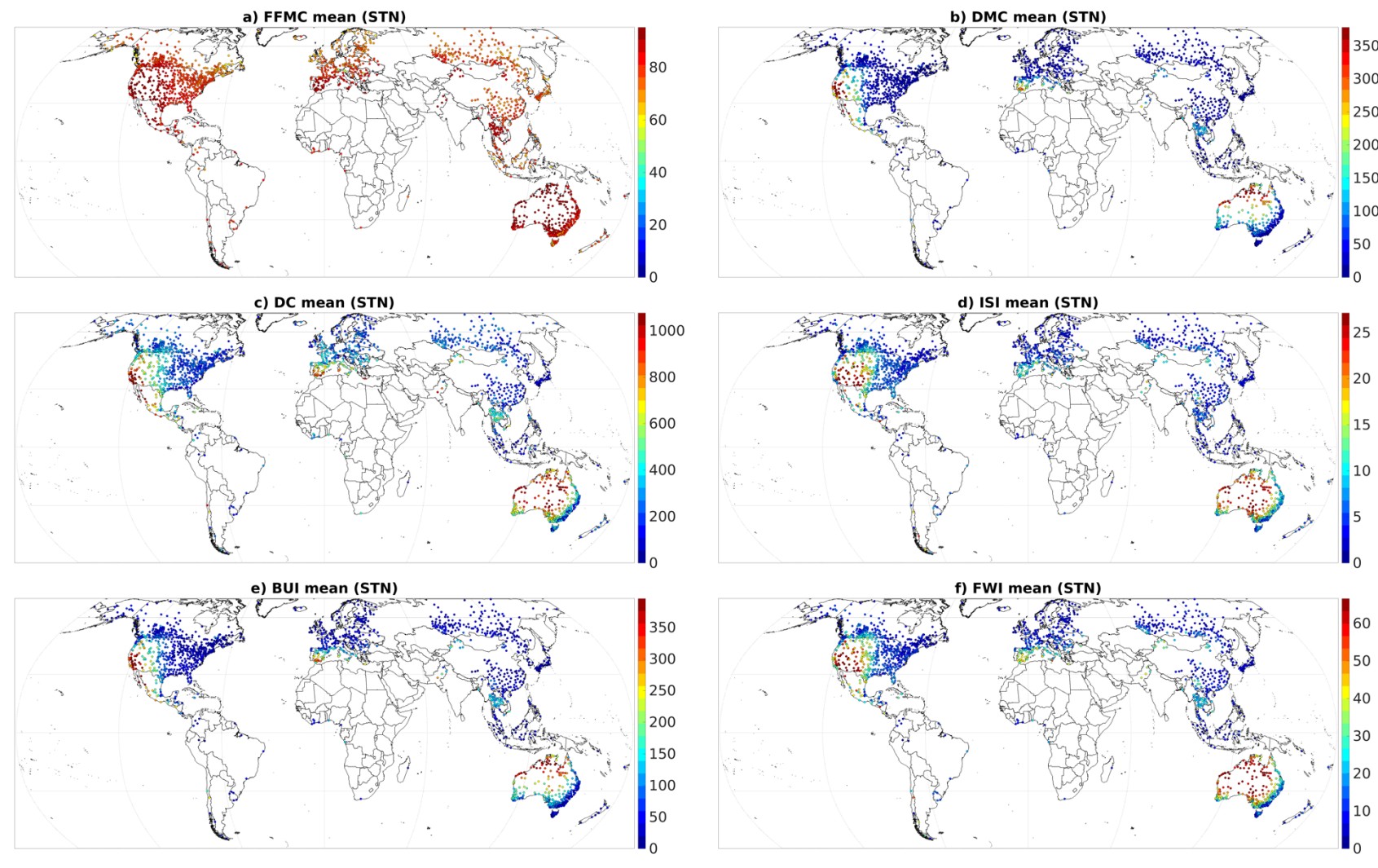

**Figure 4. 2004-2018 mean of Fire Weather Index (FWI) components calculated from weather station data, only over the local 4-month fire season.**

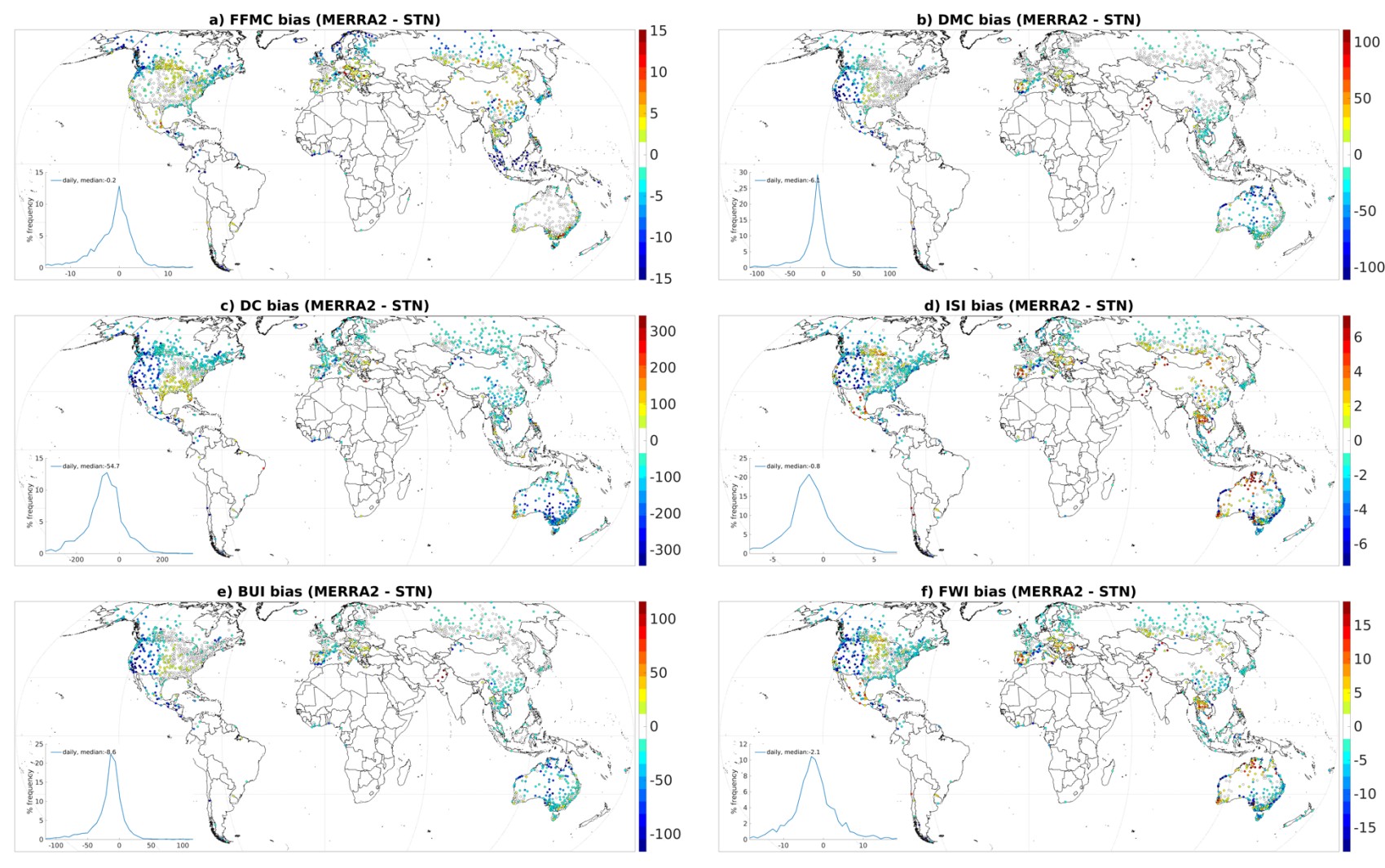

**Figure 5. 2004-2018 bias between Fire Weather Index (FWI) components calculated from MERRA2 and from weather stations, only over the local 4-month fire season.**

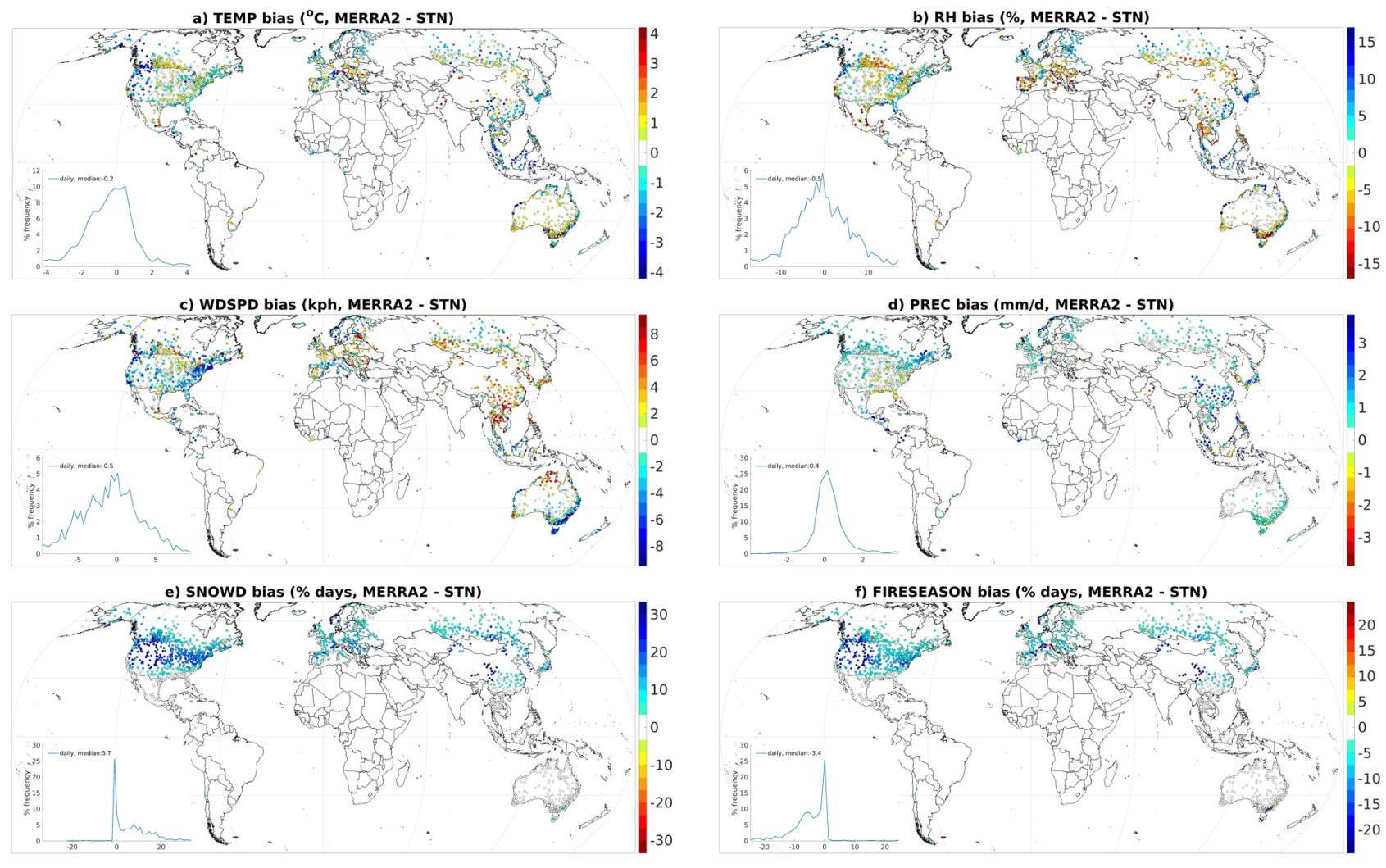

**Figure 6. Same as Figure 5 but for weather input variables. SNOWD (e) is expressed as the difference between MERRA2 and the station data in the percentage of days during the year when snow depth is greater than 1cm, the threshold below which FWI calculations are active. FIRESEASON (f) is expressed as the difference between MERRA2 and the station data in the percentage of days during the year when FWI calculations are active.**

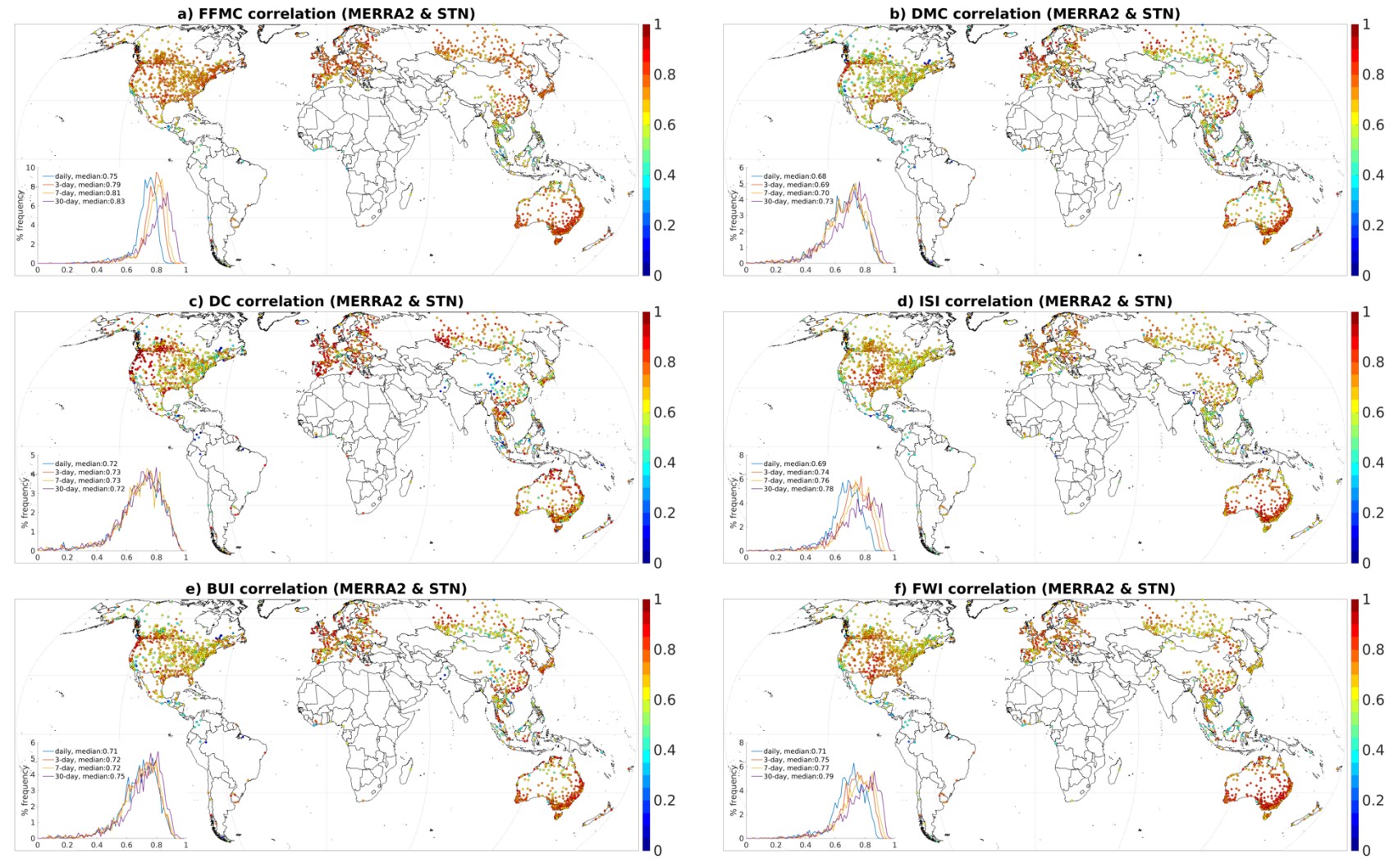

**Figure 7. Correlation between daily station and MERRA2 FWI component values over 2004-2018 during the local 4-month fire season. The inset histograms show the frequency distribution of correlations across all stations for daily, 3-day, 7-day and 30-day average FWI components.**

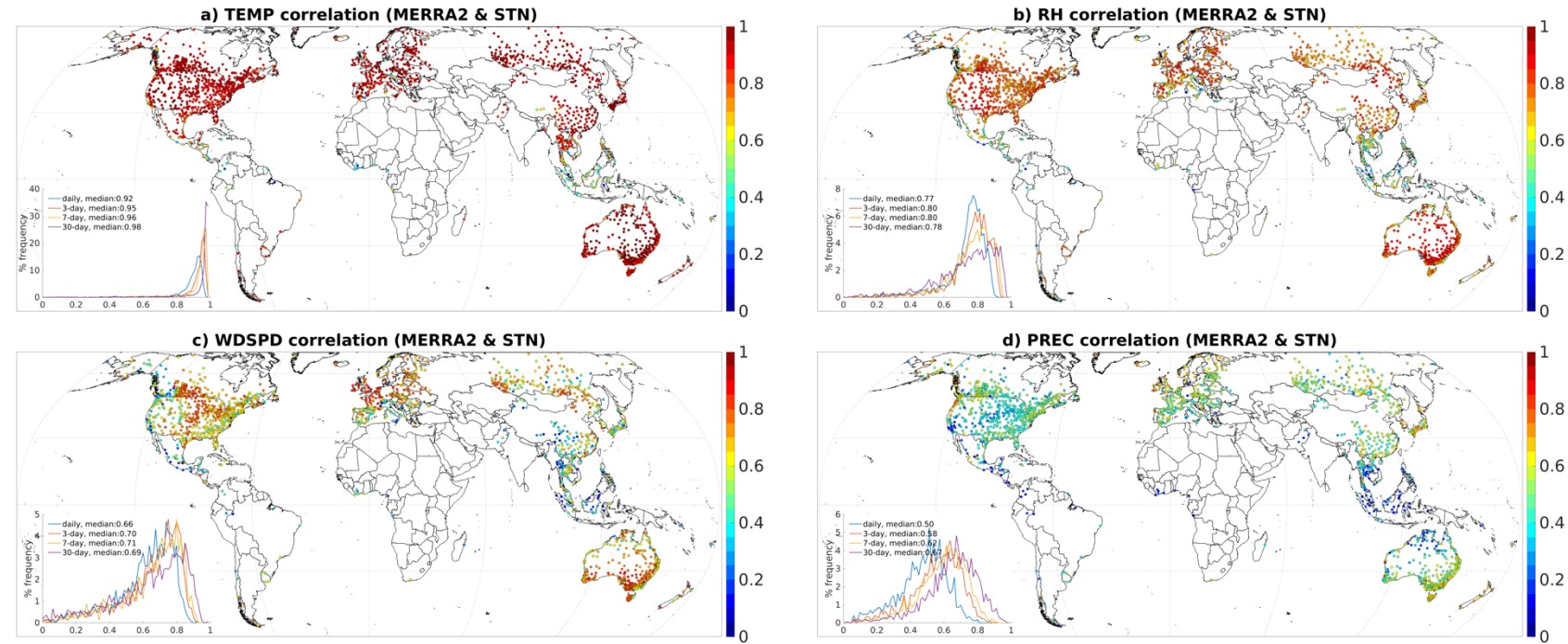

**Figure 8. Same as Figure 7, but for input weather variables.**

**Average GEOS-5 500 hPa heights (dam), 20180801 - 20180821**

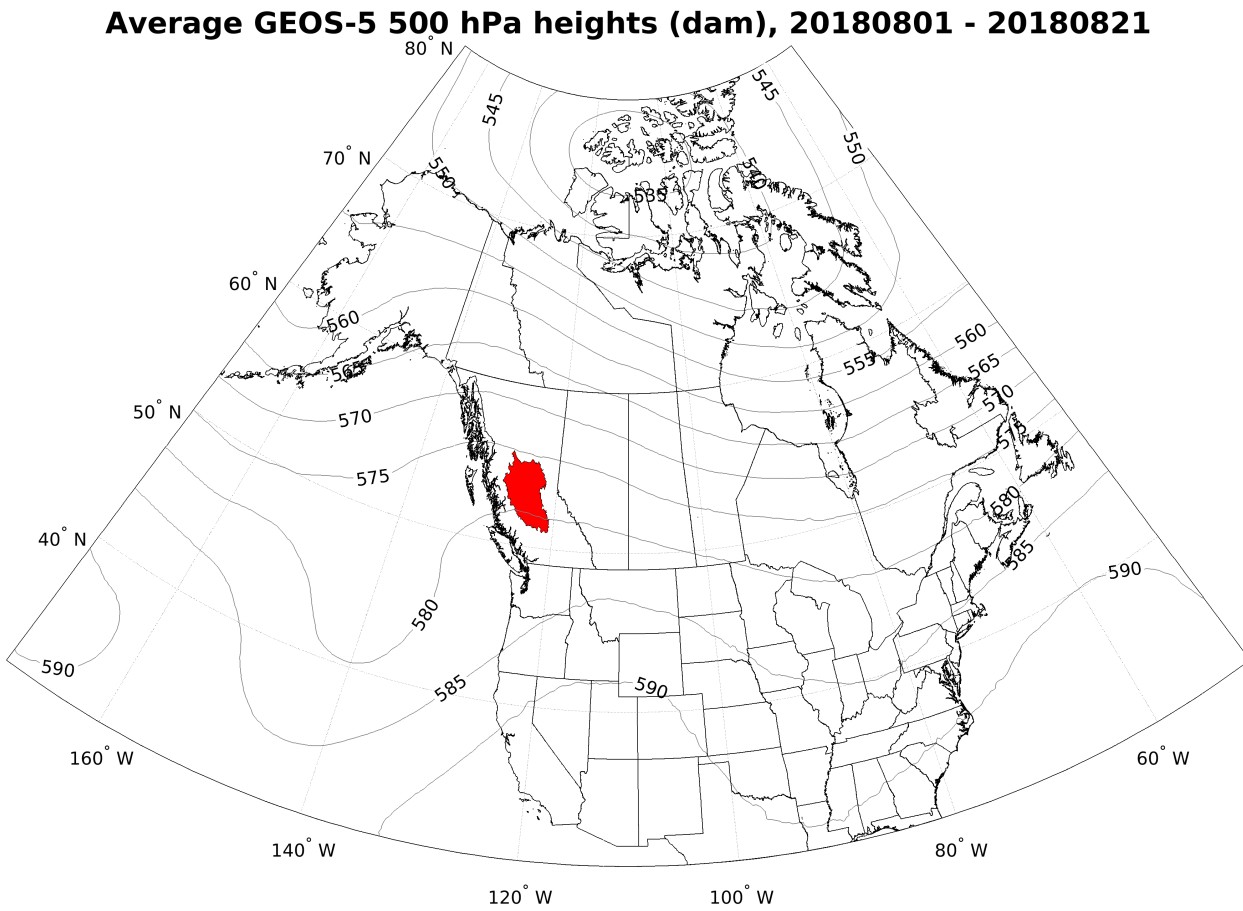

**Figure 9. Average GEOS-5 analysis 500 hPa heights (dam) over Canada and the US from August 1-August 21, 2018. The area in red is the Fraser Plateau and Basin Complex ecoregion from the Terrestrial Ecoregions of the World.**

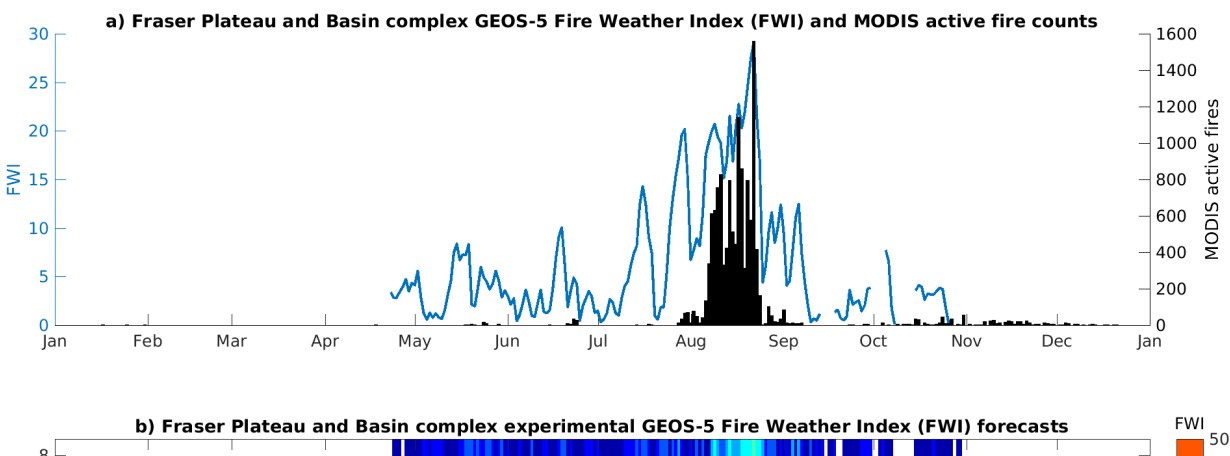

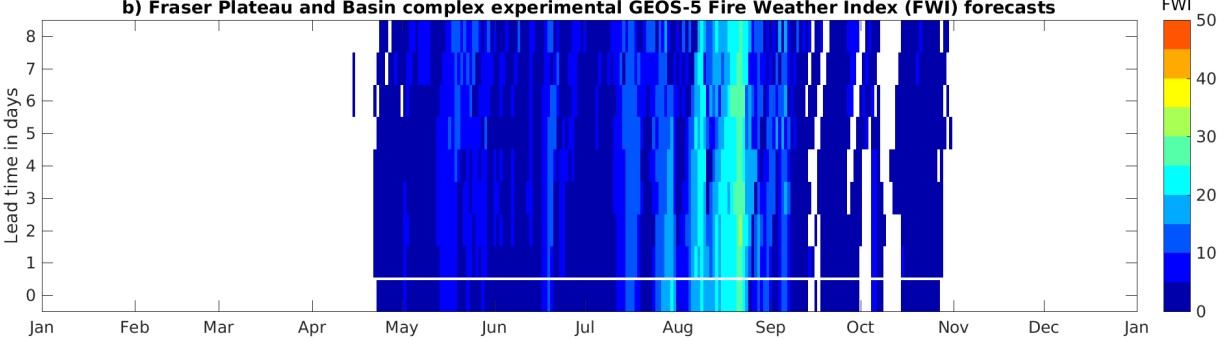

**Figure 10. a) daily MODIS active fire totals (>80% confidence only) and FWI calculated from GEOS-5 analysis field averaged over the Fraser Plateau and Basin Complex ecoregion in Figure 9. b) forecasts of the FWI at lead times of up to 8 days. The FWI in the time series of the top panel corresponds to the lead-0 row at the bottom of the coloured plot, separated from the forecasts by the white horizontal line. Missing FWI values in both panels indicate that FWI calculations have stopped due to cold temperatures or snow cover.**

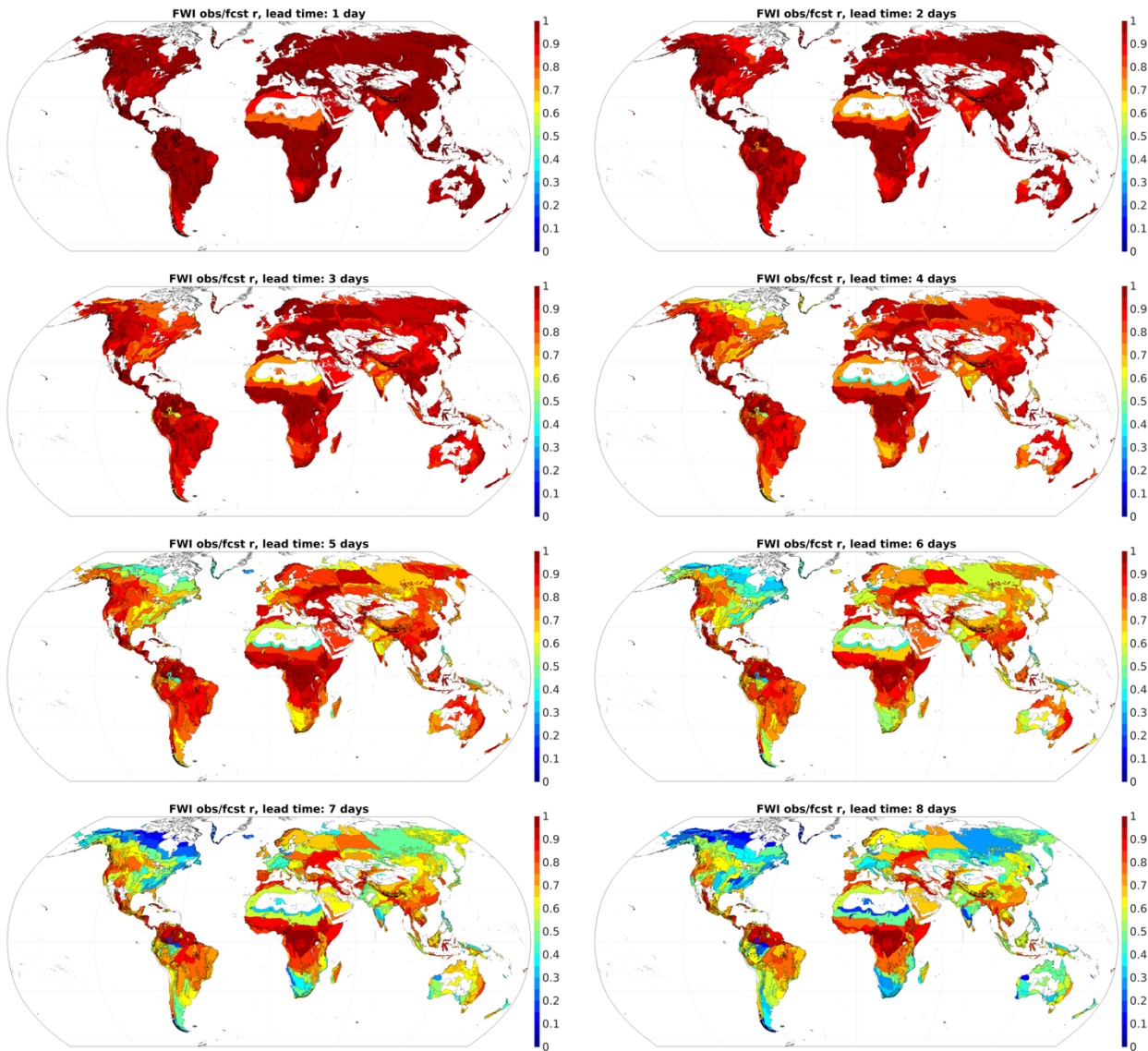

**Figure 11. Correlation (r) between daily analysis and forecast FWI for 2018 at lead times of 1 to 8 days, for GEOS-5 grid points averaged within each of 771 Terrestrial Ecoregions of the World regions. Correlations are calculated only over the local fire season in each ecoregion, defined as the four-month period with the highest mean FWI.**

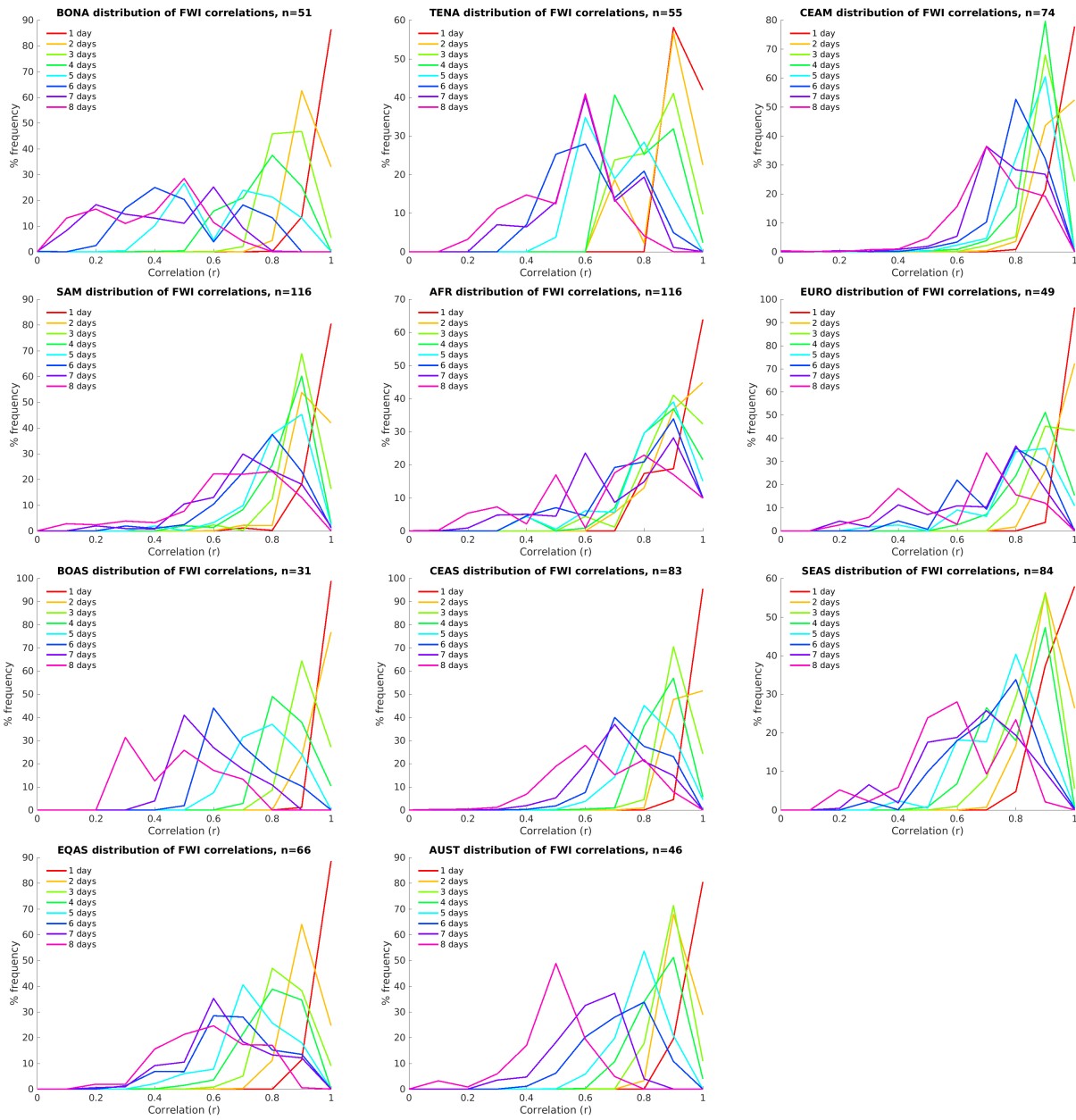

**Figure 12. Distributions of correlations between daily GEOS-5 forecast and analysis FWI at different lead times for ecoregions in each GFED region.**

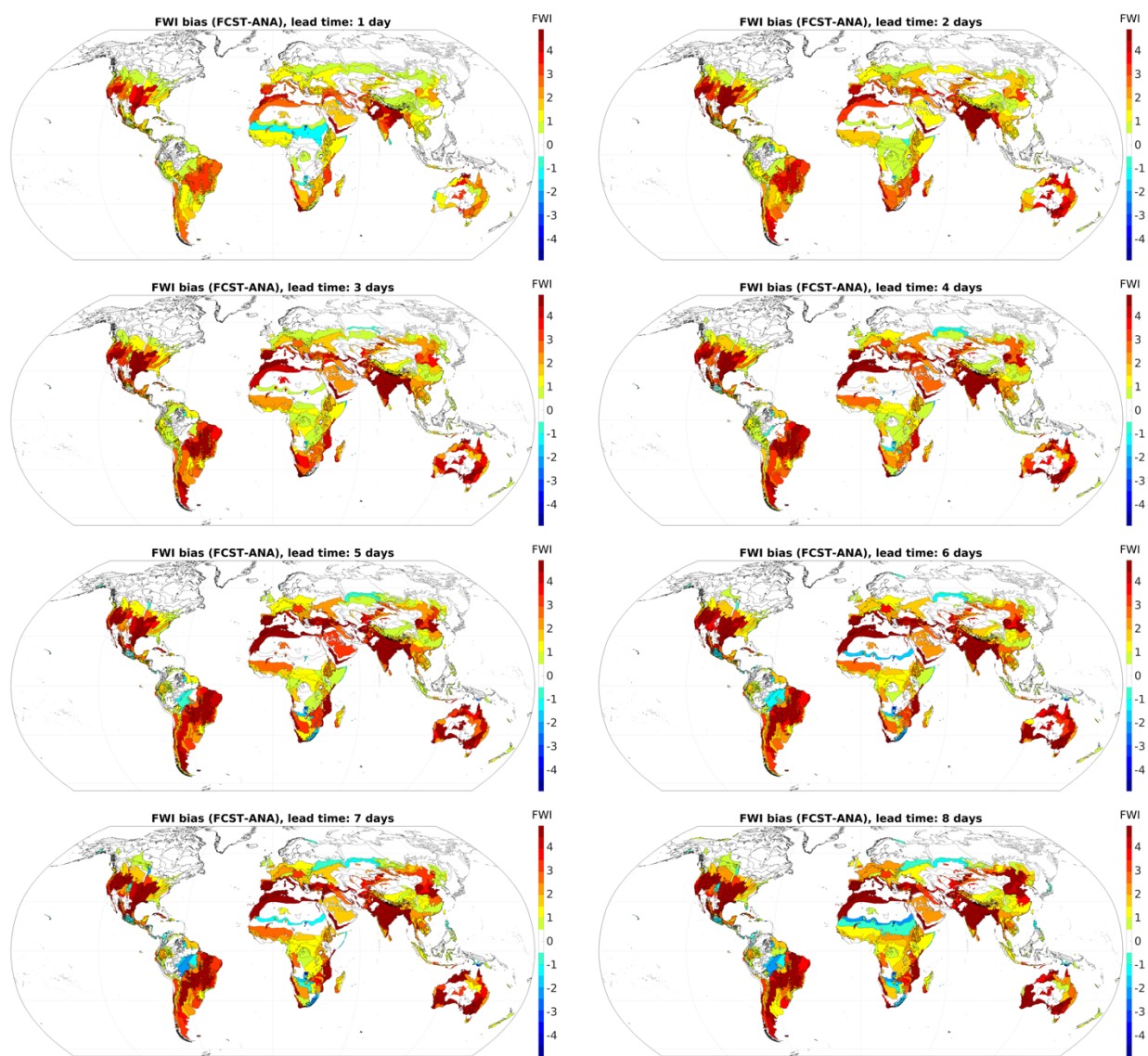

**Figure 13. Same as Figure 11, but for the FWI bias (forecast-analysis).**

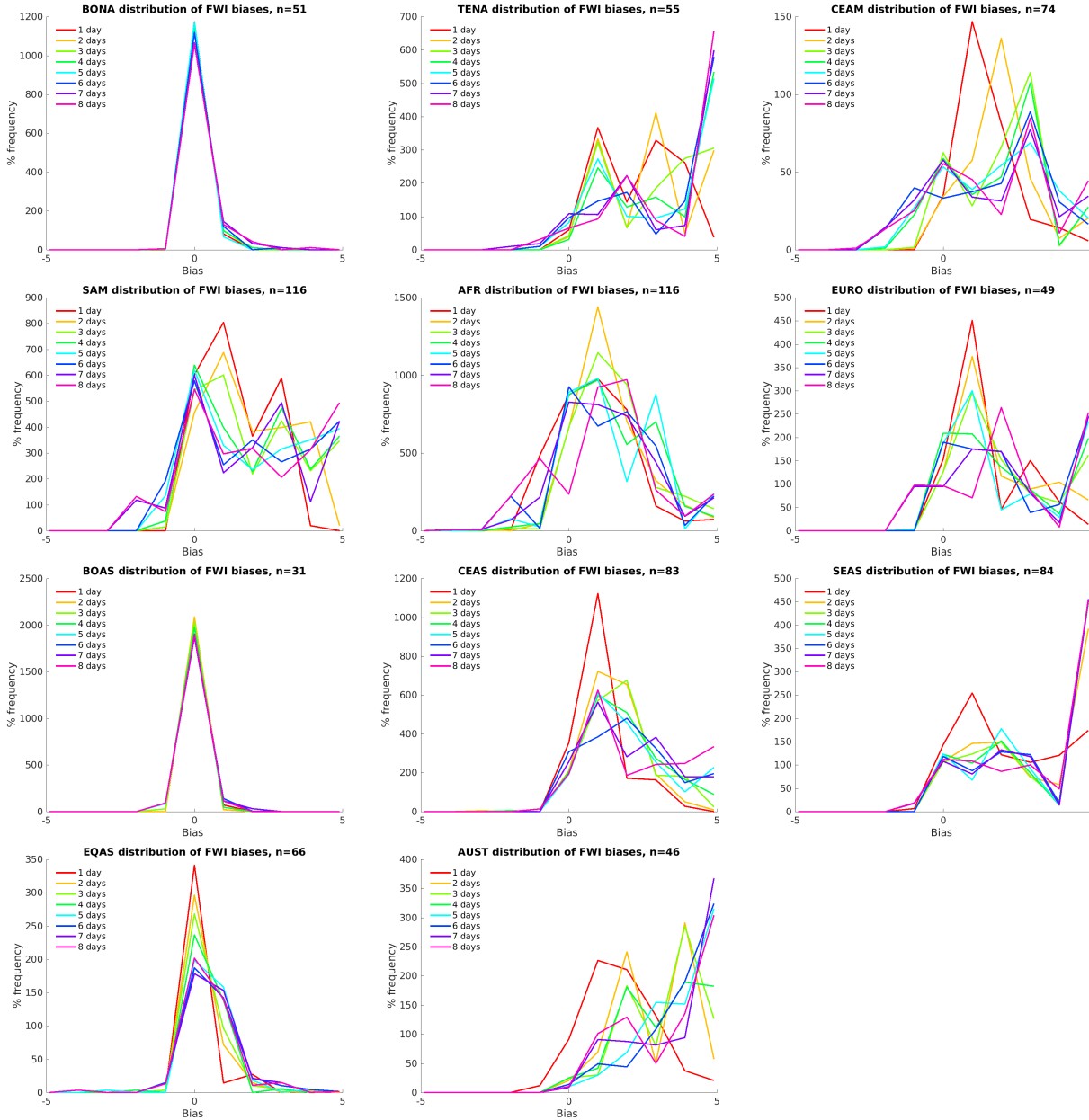

**Figure 14. Same as Figure 12, but for forecast biases.**

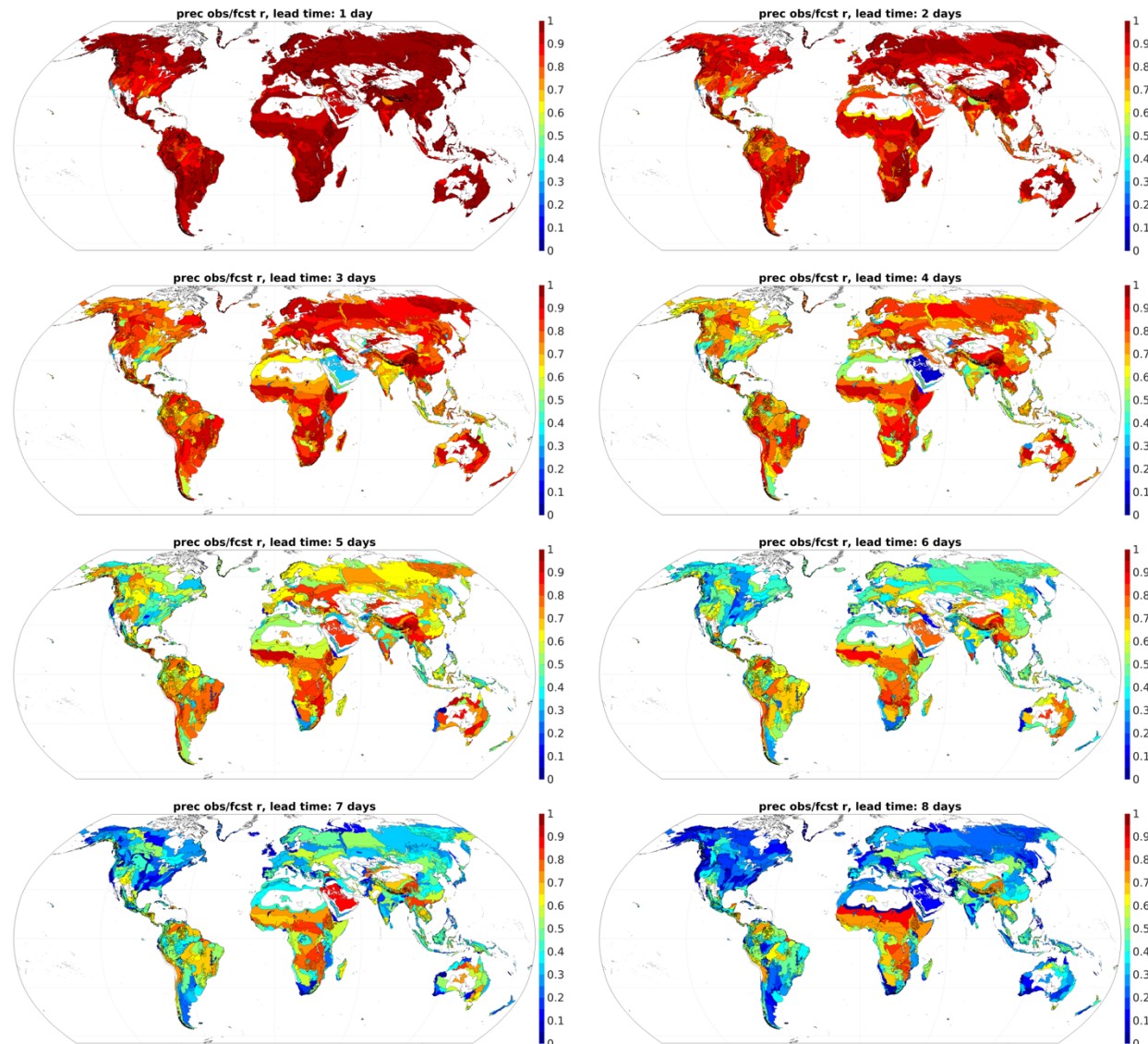

**Figure 15. Same as Figure 11, but for precipitation (PREC).**

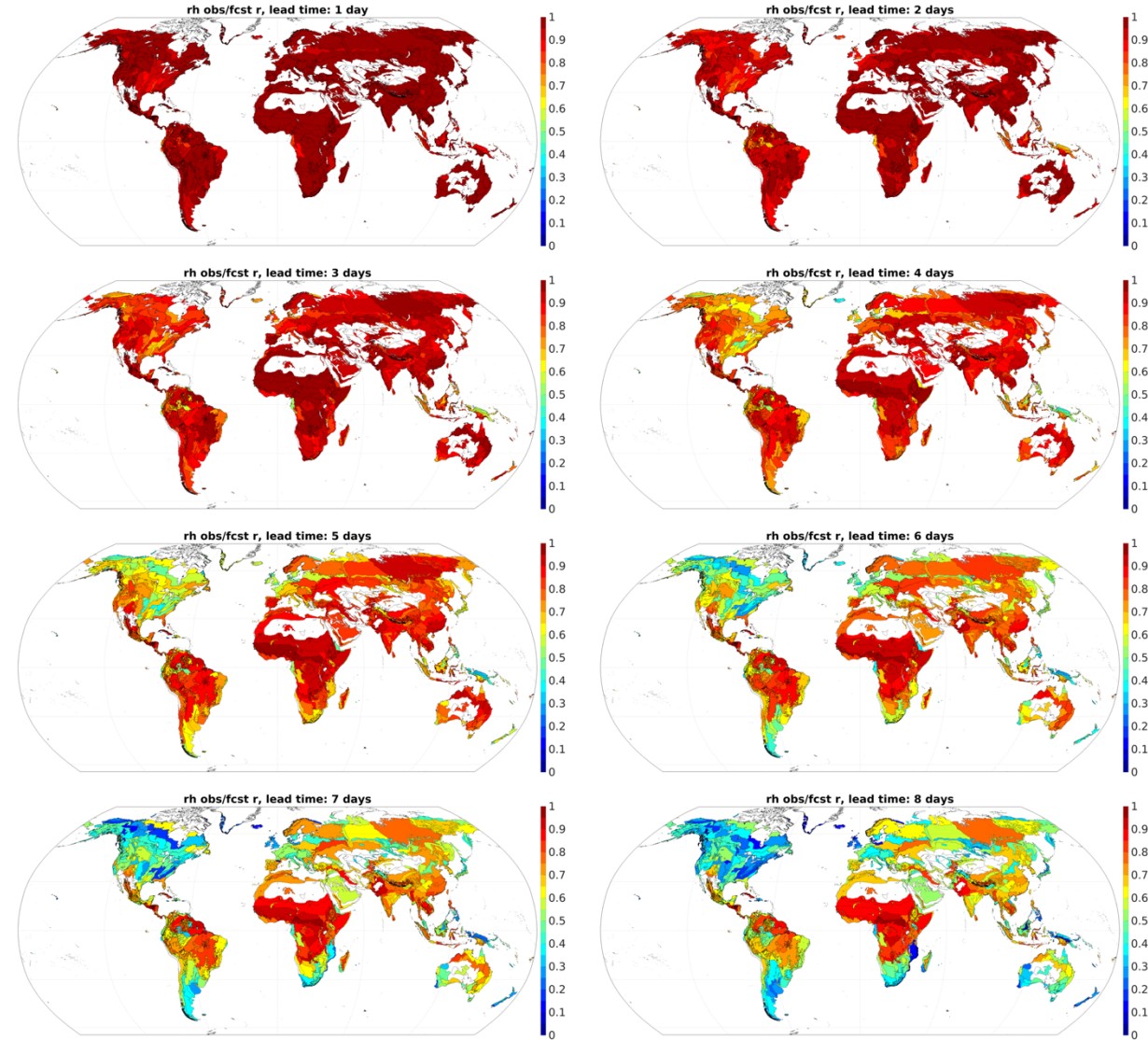

**Figure 16. Same as Figure 11, but for relative humidity (RH).**