# Peer review of "Evaluation of Global Fire Weather Database re-analysis and short-term forecast products"

_Natural Hazards and Earth System Sciences, 2019_

## Referee Comment (RC1) · Anonymous Referee #1 · 30 Jul 2019

The manuscript entitled "Evaluation of Global Fire Weather Database reanalysis and short-term forecast products" compares the FWI computed from MERRA2 reanalysis to global weather stations and evaluates the skill of FWI forecasts from NASA GEOS-5 weather forecasts up to 8 days lead time. The assessment of FWI bias concerning weather stations follows upon previous works at a regional scale and a recent global comparison with another reanalysis (ERA-Interim). The evaluation of FWI short-term forecast skill is the first at a global level, providing new insights. The manuscript read well and the overall presentation quality (structure, figures, tables, etc) is good, while the results are well described and discussed in some depth regarding several related works. As such, I believe the manuscript can be accepted with relatively minor changes as it presents a relevant contribution that is of interest to the wider fire community.

[Figure]

While the processing of the weather station data is thoughtfully described, the description of the datasets should be made more straightforward. The title of the manuscript refers to the evaluation of GFWED but, through the text, the GFWED is sporadically mentioned, being the FWI data referred to as MERRA2 FWI or GEOS-5 FWI. I am afraid this may lead to some confusion, and I suggest starting the Data and Methods section with a brief description of MERRA2, GEOS-5 and GFWED. Also, in the Data and Methods section, it is not clear what reference data is used to evaluate FWI forecast skill. Additionally, there are several references in the Introduction that are missing in the references section (e.g., references at P1 L37/38). Finally, I believe the manuscript would benefit from a conclusion section summarizing the main results. In the current form, the ending feels unexpected as if something is missing.

Specific comments P1 L21 – it should be made clear that FWI consists of three moisture codes and three fire behavior indices before describing the two groups individually; I suggest moving the description FWI inputs (P1 L40-41) to before the description of each FWI component; P8 L277 - section 4 "GEOS-5 FWI forecast evaluation for 2018" should be a subsection of results (section 3).

Technical corrections P1 L9: "NASA he Modern-Era" should read "NASA Modern-Era; P1 L24: "from" should precede "temperature"; P1 L31: "mm" is missing after "2.8". P6 L200: "FIRESEAON" should read "FIRESEASON" (the same typo appears several times throughout the text).

---

## Referee Comment (RC2) · Anonymous Referee #2 · 16 Sep 2019

**General comments**

The paper entitled "Evaluation of Global Fire Weather Database re-analysis and short-term forecast products", by R.B. Field, addresses the evaluation of reanalysis and forecast model-derived fire weather products of global coverage to provide a baseline for their application in impact studies. In particular, two main research questions are addressed:

1. All the FWI system components, as computed using the MERRA2 reanalysis, are compared against observational data from a global weather station network (n=1746) in terms of biases and their relationship with the input variables. 2. The skill of short-term FWI forecasts (up to 8-day lead time) based on the NASA GEOS-5 weather forecasting system is evaluated, considering the global observational database as ref-

erence.

This is an interesting paper, of undeniable scientific quality and relevance for the target journal. I would therefore recommend publication without major modifications.

I have elaborated a brief list of minor typos, and a few questions and suggestions. The article is well written and the results are presented with relevant tables and figures, and adequately discussed within the context of earlier research in the field. In my opinion it is a valuable contribution in the line of improving our understanding of the FWI system and to aid data users in its proper application, particularly when facing the need of using reanalysis data to this aim due to poor (or null) observational coverage, which is often the case in many impact and vulnerability assessment studies. The product here analyzed (GFWED) is of great relevance to this aim, and a thorough assessment of its advantages and limitations as compared to actual observational data is presented at a global scale. Furthermore, the assessment of the NASA GEOS-5 FWI forecast skill provides useful information for their application within operational and/or research context.

**Specific comments**

I agree with the first referee in that the manuscript would gain from a unified description of the different databases/models involved in the analysis under the Data and Methods section, so the reader can get a more straightforward overview of the data involved (GWFED, MERRA-2, GEOS-5). I also find confusing the alternative use of "GFWED" and "MERRA2 FWI" denominations throughout the text. As a suggestion to the author, it would be also interesting a very short comment on the main differences between MERRA and the newer MERRA-2 (of course, nothing too technical), and if possible to mention in a nutshell what would be the expected improvement or most relevant differences regarding the derived FWI product in both cases, apart from the citation to the work by Field et al. 2015 focused on the MERRA-based DC (L50-53).

The gaps in the input fire-weather variables TEMP, RH and WDSPD from the station
data were completed using data from MERRA2 fields at the gridboxes of each station (and precipitation from CPC records), up to 20% gaps. This is probably the least bad option in the presence of missing data, although it is obviously "favoring" the validation results at the gap-filled stations. Are the corresponding MERRA2 data being introduced directly, or is any form of bias correction being applied prior to that, so there is a smoother transition between actual records and MERRA2 values?

It is unclear what is the verifying reference against which GEOS-5 FWI forecasts have been validated (MERRA, MERRA2?). This should be made clear early in the manuscript.

In L234 some outlying values over Pakistan are mentioned. Wouldn't it be better to just discard these data with a detrimental influence on the validation results?

The results obtained indicate the need for bias-correcting the MERRA-based FWI in many real-world applications (L371-373). With this regard, it might be worth mentioning that the correction of multi-variable indices has some intrinsic complexities that, for the particular case of FWI, have been previously addressed by other authors (see e.g. Casanueva et al. 2018)

The paper contains a lot of information from the validation of GFWED and GEOS-5 forecasts. I agree with referee #1 that the manuscript would benefit from a final conclusions section summarizing the main results and conclusions.

**Technical corrections**

I have also suggested a few corrections to a few typos in the text, apart from those already indicated by the Referee #1

L24 "[. . .] is calculated temperature, relative humidity". . . Is it perhaps the word 'using' missing here?

L38 The reference to Cantin 2016 is missing in the reference list

L48 needed instead of need?

Table 2. I would suggest to include this information in the legend of Fig. 1, so this table can be eliminated.

Table 3. The columns SNOWD and FIRESEASON are well understood, but these codes have not been previously described explicitly, neither in the text, nor in the table's caption, so I would suggest to explicitly describe them prior to first using them.

L188-189 [. . .] across stations "for?" each of the GFED regions

Fig. 13. Given the wide variability of FWI magnitude across the globe, did the author consider to use here relative instead of absolute biases?

L402 Although "at" seasonal

**References**

Casanueva, A., Bedia, J., Herrera, S., Fernández, J., Gutiérrez, J.M., 2018. Direct and component-wise bias correction of multi-variate climate indices: the percentile adjustment function diagnostic tool. Climatic Change 147, 411–425. https://doi.org/10.1007/s10584-018-2167-5

---

## Author Comment (AC1) · 23 Jan 2020

**I thank the first referee for their thoughtful and specific comments on the manuscript. Point by point responses with line numbers are listed below.**

Anonymous Referee #1
The manuscript entitled "Evaluation of Global Fire Weather Database reanalysis and short-term forecast products" compares the FWI computed from MERRA2 reanalysis to global weather stations and evaluates the skill of FWI forecasts from NASA GEOS-5 weather forecasts up to 8 days lead time. The assessment of FWI bias concerning weather stations follows upon previous works at a regional scale and a recent global comparison with another reanalysis (ERA-Interim). The evaluation of FWI short-term forecast skill is the first at a global level, providing new insights. The manuscript read well and the overall presentation quality (structure, figures, tables, etc) is good, while the results are well described and discussed in some depth regarding several related works. As such, I believe the manuscript can be accepted with relatively minor changes as it presents a relevant contribution that is of interest to the wider fire community.

While the processing of the weather station data is thoughtfully described, the description of the datasets should be made more straightforward. The title of the manuscript refers to the evaluation of GFWED but, through the text, the GFWED is sporadically mentioned, being the FWI data referred to as MERRA2 FWI or GEOS-5 FWI. I am afraid this may lead to some confusion, and I suggest starting the Data and Methods section with a brief description of MERRA2, GEOS-5 and GFWED.
**L66: I have added more details on the different versions of GFWED currently available, I hope clarifying the differences between MERRA2 and GEOS-5 under the broader GFWED 'umbrella'. For completeness, I have also mentioned the satellite precipitation-based products, but because they are not included in the analysis, have left this whole description in the Introduction, rather than Data and Methods.**

Also, in the Data and Methods section, it is not clear what reference data is used to evaluate FWI forecast skill.
**L158: I have described here how the forecast FWI is evaluated against the analysis (0-day lead time). In theory, the forecasts could be compared to FWI calculated from weather stations, but the weather station coverage was simply too poor in many areas.**

Additionally, there are several references in the Introduction that are missing in the references section (e.g., references at P1 L37/38).
**Thank you for catching these. The Dowdy et al. (2009), Van Wagner (1987) and Cantin (2016) references have been added to the bibliography.**

Finally, I believe the manuscript would benefit from a conclusion section summarizing the main results. In the current form, the ending feels unexpected as if something is missing.
**Thanks for the suggestion. I have added a Conclusions section.**

Specific comments P1 L21 – it should be made clear that FWI consists of three moisture codes and three fire behavior indices before describing the two groups individually; I suggest moving the description FWI inputs (P1 L40-41) to before the description of each FWI component;

**L20: A description of the overall moisture codes and fire behavior indices has been added, and the FWI input description has been moved to the end of this paragraph.**

P8 L277 - section 4 "GEOS-5 FWI forecast evaluation for 2018" should be a subsection of results (section 3).
**Thank you for pointing this out. This now a subsection of the Results, the other sections in which have been re-numbered and renamed accordingly.**

Technical corrections
P1 L9: "NASA he Modern-Era" should read "NASA Modern-Era;
**Thank you for catching this, it has been corrected.**

 P1 L24: "from" should precede "temperature";
**Thank you for catching this, it has been corrected.**

P1 L31: "mm" is missing after "2.8".
**Thank you for catching this, it has been corrected.**

P6 L200: "FIRESEAON" should read "FIRESEASON" (the same typo appears several times throughout the text).
**Thank you again for catching these, they have been corrected.**

---

## Author Comment (AC2) · 23 Jan 2020

**I thank the second referee for their thoughtful and specific comments on the manuscript. Point by point responses with line numbers are listed below.**

Anonymous Referee #2
**General comments**
The paper entitled "Evaluation of Global Fire Weather Database re-analysis and shortterm forecast products", by R.B. Field, addresses the evaluation of reanalysis and forecast model-derived fire weather products of global coverage to provide a baseline for their application in impact studies. In particular, two main research questions are addressed:
1. All the FWI system components, as computed using the MERRA2 reanalysis, are compared against observational data from a global weather station network (n=1746) in terms of biases and their relationship with the input variables. 2. The skill of short-term FWI forecasts (up to 8-day lead time) based on the NASA GEOS-5 weather forecasting system is evaluated, considering the global observational database as reference.

This is an interesting paper, of undeniable scientific quality and relevance for the target journal. I would therefore recommend publication without major modifications.

I have elaborated a brief list of minor typos, and a few questions and suggestions. The article is well written and the results are presented with relevant tables and figures, and adequately discussed within the context of earlier research in the field. In my opinion it is a valuable contribution in the line of improving our understanding of the FWI system and to aid data users in its proper application, particularly when facing the need of using reanalysis data to this aim due to poor (or null) observational coverage, which is often the case in many impact and vulnerability assessment studies. The product here analyzed (GFWED) is of great relevance to this aim, and a thorough assessment of its advantages and limitations as compared to actual observational data is presented at a global scale. Furthermore, the assessment of the NASA GEOS-5 FWI forecast skill provides useful information for their application within operational and/or research context.
**Specific comments**
I agree with the first referee in that the manuscript would gain from a unified description of the different databases/models involved in the analysis under the Data and Methods section, so the reader can get a more straightforward overview of the data involved (GWFED, MERRA-2, GEOS-5).
**L66: Thanks for the suggestion. As described in the response to the first reviewer, this has been added in the introduction.**

I also find confusing the alternative use of "GFWED" and "MERRA2 FWI" denominations throughout the text.
**Thank you for pointing this out. To make this clearer, I have omitted mentions of GFWED wherever possible. They appear mainly in the first half of the introduction, and only MERRA2 or GEOS-5 FWI are mentioned elsewhere.**

As a suggestion to the author, it would be also interesting a very short comment on the main differences between MERRA and the newer MERRA-2 (of course, nothing too technical), and if possible to mention in a nutshell what would be the expected improvement or most relevant

differences regarding the derived FWI product in both cases, apart from the citation to the work by Field et al. 2015 focused on the MERRA-based DC (L50-53).

**L66: Thank you for the suggestion. I have added a brief summary of the changes to MERRA2 from MERRA most relevant to precipitation, which among the FWI input variables, are discussed in most detail in Gelaro et al. (2017).**

The gaps in the input fire-weather variables TEMP, RH and WDSPD from the station data were completed using data from MERRA2 fields at the gridboxes of each station (and precipitation from CPC records), up to 20% gaps. This is probably the least bad option in the presence of missing data, although it is obviously "favoring" the validation results at the gap-filled stations. Are the corresponding MERRA2 data being introduced directly, or is any form of bias correction being applied prior to that, so there is a smoother transition between actual records and MERRA2 values?

**L137: Any kind of systematic bias correction at the necessary diurnal scale was unfortunately beyond the scope of this study. To reduce the 'favoring' of gap-filled stations, I applied the further requirement beyond the initial quality screening that each daily record at each station only be included in the bias and correlation statistics if no more than 20% daily values RH (and thus also temperature) during the previous 60 days were interpolated from MERRA2 (to account for the current day's weather inputs and also antecedent weather). This was done in the original analysis and is now mentioned in the manuscript.**

It is unclear what is the verifying reference against which GEOS-5 FWI forecasts have been validated (MERRA, MERRA2?). This should be made clear early in the manuscript.

**L158: Following the same comment from the first reviewer, the GEOS-5 forecasts at 1-8 day lead time are evaluated against the GEOS-5 analysis (0-day lead) fields, which is now mentioned in the manuscript.**

In L234 some outlying values over Pakistan are mentioned. Wouldn't it be better to just discard these data with a detrimental influence on the validation results?

**L284: Thanks for the comment, that is a good point. Rather than remove the stations from the main analysis, I have mentioned in the text how the correlations over SEAS between DMC and BUI with FIRESEASON are reduced to r=-0.26 and r=-0.22 respectively when the four stations from Pakistan are omitted.**

The results obtained indicate the need for bias-correcting the MERRA-based FWI in many real-world applications (L371-373). With this regard, it might be worth mentioning that the correction of multi-variable indices has some intrinsic complexities that, for the particular case of FWI, have been previously addressed by other authors (see e.g. Casanueva et al. 2018)

**L475: Thank you for pointing out this very interesting and relevant study. We have mentioned the need for bias correction for real-world applications in the new Conclusions section, along with Yong et al. (2015) cited therein.**

The paper contains a lot of information from the validation of GFWED and GEOS5 forecasts. I agree with referee #1 that the manuscript would benefit from a final conclusions section summarizing the main results and conclusions.
**Thanks for the suggestion. I have added a Conclusions section.**

**Technical corrections**
I have also suggested a few corrections to a few typos in the text, apart from those already indicated by the Referee #1
L24 "[...] is calculated temperature, relative humidity"... Is it perhaps the word 'using' missing here?
**Thank you for catching this, it has been corrected.**

L38 The reference to Cantin 2016 is missing in the reference list
**Thank you for catching this, it has been added.**

L48 needed instead of need?
**Thank you for catching this, it has been corrected.**

Table 2. I would suggest to include this information in the legend of Fig. 1, so this table can be eliminated.
**Thank you for the suggestion. Table 2 has been removed and the GFED acronyms and descriptions have been added to the caption of Figure 1.**

Table 3. The columns SNOWD and FIRESEASON are well understood, but these codes have not been previously described explicitly, neither in the text, nor in the table's caption, so I would suggest to explicitly describe them prior to first using them.
**L224: Thank you for catching this. SNOWD and FIRESEASON have been defined at L250 in the text and in the caption of Table 2.**

L188-189 [...] across stations "for?" each of the GFED regions
**Thank you for catching this, it has been corrected.**

Fig. 13. Given the wide variability of FWI magnitude across the globe, did the author consider to use here relative instead of absolute biases?
**L217: Thanks for the suggestion. I did consider using relative biases (shown in the figure below), but decided on absolute biases for a more direct interpretation of the maps. The effect of taking the relative bias is now mentioned. The mean and biases in Tables 3-5 were included in part for more quantitative evaluation of the biases across the different GFED regions, which is now mentioned at L236.**

[Figure]

L402 Although "at" seasonal

**Thank you for catching this, it has been corrected.**

**References**

Casanueva, A., Bedia, J., Herrera, S., Fernández, J., Gutiérrez, J.M., 2018. Direct and component-wise bias correction of multi-variate climate indices: the percentile adjustment function diagnostic tool. Climatic Change 147, 411–425. https://doi.org/10.1007/s10584-018-2167-5